# Sensors that Learn: The Evolution from Taste Fingerprints to Patterns of Early Disease Detection

**DOI:** 10.3390/mi10040251

**Published:** 2019-04-16

**Authors:** Nicolaos Christodoulides, Michael P. McRae, Glennon W. Simmons, Sayli S. Modak, John T. McDevitt

**Affiliations:** Department of Biomaterials, College of Dentistry, Bioengineering Institute, New York University, New York, NY 10010, USA; nicolaoschristo19@gmail.com (N.C.); michael.mcrae@nyu.edu (M.P.M.); glennon.simmons@nyu.edu (G.W.S.); sayli.modak@nyu.edu (S.S.M.)

**Keywords:** electronic tongue, electronic taste chip, biosensors, point-of-care, biomarkers, serum, saliva, programmable bio-nano-chip (p-BNC), early disease detection

## Abstract

The McDevitt group has sustained efforts to develop a programmable sensing platform that offers advanced, multiplexed/multiclass chem-/bio-detection capabilities. This scalable chip-based platform has been optimized to service real-world biological specimens and validated for analytical performance. Fashioned as a sensor that learns, the platform can host new content for the application at hand. Identification of biomarker-based fingerprints from complex mixtures has a direct linkage to e-nose and e-tongue research. Recently, we have moved to the point of big data acquisition alongside the linkage to machine learning and artificial intelligence. Here, exciting opportunities are afforded by multiparameter sensing that mimics the sense of taste, overcoming the limitations of salty, sweet, sour, bitter, and glutamate sensing and moving into fingerprints of health and wellness. This article summarizes developments related to the electronic taste chip system evolving into a platform that digitizes biology and affords clinical decision support tools. A dynamic body of literature and key review articles that have contributed to the shaping of these activities are also highlighted. This fully integrated sensor promises more rapid transition of biomarker panels into wide-spread clinical practice yielding valuable new insights into health diagnostics, benefiting early disease detection.

## 1. Introduction

Biomarker measurements are increasingly essential for assessing patient health and guiding clinical practice. Yet despite the ubiquity of biomarker data in academic research, the clinical translation rate of biomarkers has remained stagnant. For instance, between 1995 and 2005, there were >26,000 publications for both cancers and cardiovascular diseases; however, the Food and Drug Administration (FDA) has approved only ~1 protein biomarker per year [1]. This gap in translating academic/commercial biomarker efforts to widespread clinical use may be bridged by new technologies, such as the programmable Bio-Nano-Chip (p-BNC) developed by the McDevitt group and reviewed here.

Another challenge that remains unresolved for the laboratory testing community is the lack of a suitable platform on which high-fidelity multiplexed and multiclass assays may be completed.

Inspired by the principles of microelectronics, such as those of scalability and microfabrication [2,3], sensors based on micro-electro-mechanical systems (MEMS) are becoming increasingly attractive for the completion of chemical (chem-) and biological (bio-) assays. The related field of the lab-on-chip (LOC) places strong focus on the integration and miniaturization of lab functions into chip structures allowing for the completion of steps more traditionally completed in the laboratory. The MEMS, LOC and microfluidics disciplines that underlie the activities of this body of work involve cross-disciplinary experience from engineering, chemistry, physics, and medicine.

Analogous to the early days of computers, when computers were too bulky to be used outside a dedicated room and require an expert operator for its use, today most of the bioanalytical data is acquired by high volume, high-throughput clinical analyzers operated by experts. Obviously, this specialized remote laboratory centered approach has significant scalability gaps required to solve many current medical problems. Gaps here include the inaccessibility of affordable healthcare by a growing population, the ever-increasing need for quick and accurate diagnostic information, and a need for reliable diagnostic solutions in developing nations.

The prospect of delivering personalized healthcare decisions at the point of care (POC) is highly attractive to physicians and their patients. Key to the delivery of this near patient testing is the time course of the measurement. New technologies that service the POC environment must be able to capture, with high performance, the key biomarker panels in a time frame that is less than the typical doctor’s office visit of 30 min or less. The POC platforms have multiple advantages over central laboratory techniques, such as non- or minimally invasive sampling techniques, smaller sample and reagent volume requirements, faster time to results, and a high degree of integration [4]. Furthermore, POC devices are usually compact and more affordable, thereby making them attractive for applications beyond the reach of central laboratory facilities [5].

The mission of the McDevitt research team is to have a significant impact on healthcare through early detection of disease and, likewise, to reduce healthcare costs and to save lives. To be successful in this mission the following three goals will be addressed:(1)To develop and integrate LOC methods for general chemistry, proteomic, genomic, and cellular analyses.(2)To accelerate the translation of laboratory findings into point of need sensors for real-world applications; that is, moving technologies from bench-to-bedside.(3)To create an efficient pipeline of medical diagnostic devices hosting relevant tests to measure condition-specific relevant biomarkers and identification of disease-specific fingerprints.

Likewise, the long-standing hypothesis of the McDevitt research team is that biomarker fingerprints might be used for early disease detection. This form of sensing of fingerprint combination from complex mixture has a direct linkage to e-nose and e-tongue research, although the original applications in these biometric fields focused more specifically on profiling smell and taste.

Our sustaining efforts of using algorithms to define health conditions has recently moved into establishing a programmable platform that can readily add new content as required for the application at hand. This work over the past two decades has moved to the point of big data acquisition alongside the linkage to machine learning and artificial intelligence. These activities point to the importance of sustained efforts to develop new technologies that impact clinical care, the translation journey that so often remains incomplete. This article describes how the electronic taste chip system evolved through a p-BNC system into a platform that digitizes biology and ultimately into clinical decision support tools. These efforts are completed within the context of a dynamic body of literature that contributes to the shaping of the activities. These complementary activities are described in some detail and key review articles are mentioned for additional coverage.

Overall this article highlights the exciting opportunities afforded by multiparameter sensing that mimics the sense of taste, but moves past the limitations of salt, sweet, sour, bitter, and glutamate into the digital fingerprints of health and wellness. Likewise, this review article highlights two decades of evolution of a platform, from an ‘electronic taste chip’ that originally focused on tastant-related chemistries (i.e., electrolyte, metal ions, sugars, and toxins) and ultimately moving into biosensing themes where it has been validated as the p-BNC. More recently, the approach has been expanded above and beyond soluble chemistries into cell counting and single cell profiling creating a ‘platform to digitize biology’. Collectively, these efforts define a pathway to ‘sensors that learn’.

## 2. Sense of Taste

Most of what is known about taste is learned through experience, whereby taste patterns and preferences are acquired. Few taste preferences are biologically preset. For example, as infants we are hard-wired with the perception that sweet is good and bitter is bad. In contrast, most taste preferences are linked with experiences. Similarities in taste preferences among different people usually are a reflection of similar experiences with types of flavors and foods. However, genetic factors may explain differences in taste perception. Taste preferences are first initiated in the womb and continue to evolve during our entire lives. The average number of taste buds in the human tongue averages between 2000 and 8000. Taste buds begin to appear during the first eight weeks of gestation; around the 12th week of gestation, when the fetus starts swallowing, the fetal taste receptors are stimulated by aroma compounds in the amniotic fluid [6]. Various nuclei of the brain stem receive taste stimuli, thereby inducing salivary flow reflexes and tongue movements. Connections between stimulated taste receptors and facial expression reflexes are observed during weeks 26 to 28 of gestation, especially with bitter taste stimuli. A couple of weeks later, the fetus changes its drinking behavior in response to a change in amniotic fluid taste. The swallowing pattern of the fetus is adjusted to a higher or lower frequency depending on the sweetness or bitterness of the amniotic fluid, respectively [6].

When a flavor or food is accepted, future preferences are likewise influenced. Interestingly, this flavor-learning has the implication that new foods eaten in conjunction with familiar foods are more likely to be accepted. Even though this effect is predominant with negative taste stimuli [7], taste stimuli may positively reinforce future individual preference for various foods [8].

The senses of taste and smell have been vigorously pursued and translated into concepts of electronic sensors, as described in the following sections.

## 3. Background on Electronic Nose vs. Electronic Tongue

Olfaction, or sense of smell, is both highly sensitive and highly discriminatory for odors. In a seminal paper published in 1982, Persaud and Dodd suggested that to discriminate complex odorant mixtures with varying odorant ratios precisely in the absence of highly specialized peripheral receptors, the olfactory systems use broadly tuned receptor cells organized in a convergent neuron pathway [9]. This research team constructed an ‘electronic nose’ using semiconductor transducers and incorporating design features so as to look into this hypothesis. They reported that discrimination between a wide variety of odors with this type of device is reproducible and that discrimination in an olfactory system is achievable without highly specific receptors.

Since the initial introduction several excellent reviews [10,11,12,13,14,15,16,17,18,19,20], as well as a plethora of very interesting research papers, have been published on the topics of electronic nose and electronic tongue [21,22,23,24,25], including electronic tongue sensing for wine [21,22], water analysis [23], tea sensing [24], and chemical sensing [25].

Around the mid-1990s exciting new developments allowed the expansion of sensor methodologies, supported by polymers creating sensor functioning as ‘electronic noses’ [26,27,28,29].

Analogous to the sense of smell, artificial nose systems assayed various vapor-phase analytes by applying a variety of detection modalities for the analyses of vapor-phase reagents, including conductive polymers [30], carbon black insulating polymer composite [31], dye-doped polymer matrixes [32], modified tin oxide sensors [33,34], quartz crystal microbalances [35], and surface acoustic wave transducers [36].

A chemo-selective colorimetric vapor sensor array then emerged that produced detectable spectral changes as a result of the attachment of volatile ligands (in vapor-phase) to immobilized metalloporphyrins [37].

These aforementioned works demonstrated the utility of array systems to analyze complex vapors, these structures were limited to the vapor phase. The extension to liquid phase complex fluid measurement is key, given that the vast majority of biological processes occur in the liquid phase. Thus, there was a compelling need to develop solution phase multianalyte detection system.

Based on the International Union of Pure and Applied Chemistry Analytical Chemistry Division (IUPAC) definition, “the electronic tongue is a multi-sensor system, which consists of several low-selective sensors and uses advanced mathematical procedures for signal processing based on the pattern recognition… and/or multivariate analysis” [38]. Sensor stability and cross-sensitivity (i.e., a reproducible response to many different species) are characteristics of critical importance. The capability of the electronic tongue to recognize qualitatively and quantitatively components of multispecies solutions of different natures is dependent on whether the sensor is properly configured and trained (i.e., calibrated) [38]. The taste sensor developed by Toko et al. is considered an electronic tongue with global selectivity (defined as “the decomposition of the characteristics of a chemical substance into those of each type of taste and their quantification”). This sensor included multichannel electrodes using a lipid/polymer membrane as the transducer [39,40,41,42,43,44,45,46]. This taste sensor was commercialized in the SA 402B and TS-5000Z systems, which are the original commercial electronic tongue systems capable of discriminating and quantifying tastes [40,41,47,48,49].

Meanwhile, the electronic tongue proposed in 1995 is defined as “a sensor used to analyze solutions using the arrays of nonspecific chemical sensors and pattern recognition” [38,50,51,52]. These academic efforts have led to commercialized electronic tongue and taste sensors, such as the Astree II e-tongue Sensor (Alpha MOS, France) used to discriminate solution samples, and the SA 402B and TS-5000Z (Intelligent Sensor Technology Inc., Japan), taste sensing systems mainly used to quantify the intensity of each type of taste identified by the human tongue using a taste ‘scale’ [53,54,55].

Dr. David R. Walt has made significant contributions in this area, also. As one of the pioneers of fiber optic-based chemical detection and biosensors, his work originated with photopolymerization at the distal end of fiber optics in 1998 [30]. Following the initial demonstration of fiber optic-based sensor, his group introduced the concept of etch pits in optical fibers, work that allowed for the expansion to a broad range of chemistries, including optical fiber arrays for DNA analysis [56,57,58,59,60,61,62]. First, single-stranded DNA is coupled to bead microspheres. Different DNA sequences are attached to different batches of beads. The beads are combined and loaded into the arrays. Dyes to encode the various bead types, and the bead positions can be identified by color. Alternatively, the arrays can be decoded using the intrinsic sequence information after a series of hybridization reactions [62]. This work has implications on lab-based application of genome sequencing and high-performance protein detection using single molecule detection strategies [63].

## 4. Electronic Taste Chip

Around the same time as the Walt etched fiber work, a new sensor methodology, described as an ‘electronic taste chip”, was introduced by the Anslyn, Niekirk, Shear, and McDevitt laboratories [64,65,66]. These initial efforts were also inspired by the human sense of taste and included use of chemically sensitive microbeads with function of ‘artificial taste buds’ that populated anisotropically etched silicon wafers. The strategic focus on scalable approaches to fabricate chip-like structures provided a direct pathway to integrate flow through microfluidic handling capabilities that greatly simplify the transport of samples and reagents as required to complete high performance assays. This initial choice of engineering platform has strong implications on the efficiency, cost, and time-scale of the platform that will ultimately lead to key considerations on scalability of these test ensembles as described in more detail below. Our efforts focus on engineering choices that have the potential to lead to ubiquitous sensing in remote locations free from the limitations associated with remote lab operations. Near patient testing, home monitoring, and testing within pharmacies are key domains that were envisioned by the initial choice of platform architecture. Another key consideration here is the choice of sample size and the definition of analysis time. Drops of bodily fluids to be measured during the time course of a doctor’s visit (i.e., less than 30 min) were defined as the ‘sweet spot’ of our choice of engineered solutions. To be clear, these choices may be considered restrictive in terms of the ultimate performance, but with proper choices of sensing strategy it is possible to provide unique sensing capabilities never seen to date for point of need sensing (see below).

The instrumentation and sample requirements for lab-based testing versus near patient testing plus the goal of moving a universal sensing platform into global settings have placed restrictions on the platform that we believe are key to moving these technologies into global healthcare settings, one of the ultimate goals for these efforts. Throughout this manuscript, there is a strong focus on the development of engineering solutions that are applied to chem- and bio-sensing paradigms that scale. The ability to capture a myriad of biomarkers with high performance on a short time scale consistent with direct feedback to the consumer is key to these choices of sensor technology.

It should be noted that both the McDevitt and Walt labs have developed sensing modalities that use bead elements as the location for the analyte detection. These efforts diverge significantly in terms of the ultimate form factor and scalability implementations, as well as the use of surface versus volume sensing elements. The Walt efforts have spun off into Illumina and Quanterix commercialization efforts, both of which are focused on larger scale instruments with high performance suitable for remote lab testing. For example, Quanterix provides a floor-standing instrument, the Simoa HD-1 Analyzer, with dimensions 141 cm × 79 cm × 161 cm. The dimensions of SensoDx instrumentation spun out of the McDevitt lab are significantly smaller (22 cm × 22 cm × 30 cm), thereby occupying a significantly lower volume (less than 1%) than the Quanterix instrument, consistent with the goal of ubiquitous sensing.

Returning to the core technology developed by the McDevitt lab, the combination of techniques and efficient/rapid data acquisition using a charge-coupled device (CCD) with known chemical indicators led to the development of a single sensor suite, described as a primitive ‘electronic taste chip’ (Figure 1). More specifically, poly(ethylene glycol)-polystyrene (PEG-PS) resin beads were derivatized with a variety of indicator molecules and were exploited to mimic ‘taste buds’ [66]. These indicators, though selective for individual analytes, were not highly specific as recognition elements. The resin beads were positioned within micromachined wells formed in Si/SiN wafers, analogous to the cavities in which natural taste buds reside, thus confining the beads to individually addressable positions on a multicomponent chip. Please note that the bead sensor ensemble used here is chemically accessible to the core. That is, they are volume-based sensing elements where signal develops through the entire volume of the bead. While there is a polymer backbone for attachment of signaling and binding agents, the vast majority of the internal volume is occupied by liquid and thus can be used for signal generation. The integration with microfluidic elements allows for rapid and efficient introduction of samples as will be described in more detail below.

Absorption properties of beads upon exposure to analytes using a CCD that was interfaced with the sensor array were used to accomplish signal transduction. Red, green, and blue (RGB) light intensities were acquired for individual beads using a CCD as the data acquisition vehicle (Figure 2). The resulting patterns were analyzed for analyte identification and ultimate quantification.

This process was indeed the first demonstration of a sensor utilizing an array of addressable bead sensors of taste and microfluidics, a potentially completely integrated system that is also programmable and scalable.

Analogous to the sense of smell, the sense of taste results from complex chemical analyses completed in a parallel fashion at the series of chemically active sites (i.e., the taste buds), located within depressions in the tongue where the molecular and ionic analytes become restricted to allow time for their identification [67,68]. A menu of chemistries has been adapted to this bead sensing element to cover the 5 taste groups, including: sour (acidity), sweet (carbohydrate-based), salty (ionic), bitter (quinine and other alkaloids), and glutamate (savory). In nature, these taste groups are used to analyze the foods and beverages. The combination of the magnitude of these signals with nose-derived information lead to a distinctive pattern for each tastant (Figure 3).

## 5. Moving Past Taste and into a Programmable Sensing Platform

While the initial implementation of the electronic taste chip allows for the simultaneous measurement of a broad range of chemistries, the linkage to biomolecules affords a series of changes in approach that are required to move in the direction of completing competitive bio-assays. This expansion of the ‘palate’ for the electronic taste chip creates an immediate need for an ever-expanding menu of biomarker derived tests. Figure 4 summarizes a partial range of applications that has been made possible with the implementation of the p-BNC approach. To date, all of these sectors have been demonstrated. Representative examples will be described below. Furthermore, an additional requirement here introduced is the capacity to customize assay conditions through a platform that is flexible, agile, and programmable. With this expansion in focus, the terminology of p-BNC is introduced [69,70].

In making the shift to the p-BNC system, the imaging approach was changed from absorbance to fluorescence measurements. In addition, the PEG bead sensor ensembles are replaced by agarose, driven by the desire to create a test micro-environment that is resistant to the problems on nonspecific binding. Agarose is derived from polysaccharide backbone that is nonimmunogenic and, as such, the matrix is well suited for high performance bio-assays. In addition to the changes in imaging mode and the bead matrix, steps were taken to upgrade the level of integration of the instrumentation en route to developing a programmable assay platform suitable for rapid optimization of bio-assays.

With this programmable platform, the initial application of chip-based multianalyte detection systems for cardiac applications was completed by targeting two inflammatory biomarkers, C-reactive protein (CRP) and interleukin-6 (IL-6), both relevant to the cardiac disease cascade [71]. From these initial efforts, it was established that Alexa fluor-488 serves as a fluorophore superior, both in terms of signal intensity and stability relative to the vast majority of other organic and inorganic fluorophores (i.e., quantum dots) [72].

For selected bioassay applications for areas like biodefense monitoring, where most situations lead to absence of signal, it was useful to develop efficient methods for the recycling and reuse of the arrays. It was determined that most efficient release of the sequestered CRP was accomplished with elution solutions of 0.1 M glycine-HCl (pH 2.5) and MgCl2 (pH 1.2). The effectiveness of the recycling method based on this treatment was shown in the colorimetric mode with three successive repetitive CRP assays without a significant loss in sensitivity [71]. Alternatively, more than 10 repetitive CRP assays were achieved easily with MgCl_2_ as the recycling agent when the assay was performed in fluorescence mode, utilizing a fluorescently conjugated CRP-detecting antibody [71].

Through the years, we have taken advantage of the unique features of the p-BNC platform to develop superior immunoassays relative to enzyme-linked immunosorbent assay with the most wide and sensitive detection capabilities, both in singleplex as well as in multiplex formats. The bead-based assays of the p-BNC undergo a vigorous optimization and validation process to ensure the assays are specific, sensitive, accurate, and precise. Typical results are shown in Figure 5 and Figure 6 developed for the cardiac application and in Figure 7 for ovarian cancer. Assay performance in terms of limit of detection, assay range, and assay precision are found on the p-BNC to be superior to the more traditional standard ELISA-based approaches.

While agarose beads had been extensively used, studied and reported on previously for areas like protein purification, our efforts revealed that robust single bead sensing could be completed using spheres derived from this matrix. With these successful bio-assays it became useful next to model the convection, diffusion, and binding kinetics of soluble reagents captured within such fibrous networks. Three-dimensional computational modeling of the capture and detection of representative protein and genetic biomolecules in 290 μm porous beads provided initial evidence for its agreement with experimental results as well as providing key insights about the process [74,75,76]. This model compared antibody-mediated capture of CRP and bovine serum albumin (BSA), along with hybridization of oligonucleotide sequences to DNA probes. The results suggested analyte transport takes place both by diffusion and convection, which is attributable to the porous nature of the interior of the bead. Furthermore, independent of the nature of analyte, the bead interiors revealed an interesting trickle of convection-driven internal flow (Figure 8). According to this model, beads with agarose concentration ranging from 0.5–8% for the sensor ensembles studied exhibited internal to external flow rate ratio between 1:3100 and 1:170 (Figure 9). Experimental evidence was in agreement with the model-derived evidence that binding kinetics strongly affect analyte distribution of captured reagents within the beads (Figure 10). These findings also revealed that high association constants lead to a boundary in which unbound analytes are retained at the periphery of the bead sensor. Furthermore, low association constants create a shallow moving boundary in which unbound analytes diffuse further into the bead before binding. The agreement between models and the experimental evidence confirmed that this new tool would be very useful for investigations of bio-agent transport taking place within these bead-based microdevices.

Functionalization of hydrogel with various capture probes and hydrophilic nano threads within the porous matrix has been used for detection of proteins [77], nucleotides [78], and cells [79]. It has been reported that improved protein activity is retained with the hydrophilic surfaces of these fibers over that of planar sensor structures [80]. The electron microscopy image of Figure 9A reveals the fiber morphology of a homogenous agarose bead synthesized with emulsion polymerization, while Figure 9B shows the structure of a superporous bead. While on first impression, the surface of the homogenous bead seems smooth, when magnified, as shown in Figure 9C, the densely packed nanofibers of the homogenous beads exhibit pore features of about 100–200 nm, consistent with microscopy within non-spherical, porous matrix [69,70]. The superporous beads, developed by a procedure known as “emulsions of emulsions” or double emulsions (complex systems whereby dispersed phase droplets also include one or more types of smaller dispersed droplets) demonstrated macropores that form interconnected compartments within the spherical ensemble. When non-cavity segments are magnified, it is revealed that pore sizes are similar to those exhibited in the homogenous beads (Figure 9D). Furthermore, because of the 3-dimensionality of the bead ensemble, the signal is aggregated over a large number of regions. For example, signal from the agarose element is derived from a thickness in a range 1000–20,000x greater than that of monolayer from a typical ELISA plate [81].

Superporous sensor ensembles have been shown to exhibit enhanced analyte transport for both cells and proteins into internal bead structures [82,83,84,85]. For 100–800 nm sized beads, as in the case of homogenous spheres, superporous beads possess ultra-large flow cavities 10–30 µm in diameter and afford rapid fluid penetration into bead center. Shorter equilibrium periods are observed relative to homogenous counterparts.

The intra-particle fluid active transport in superporous beads as reported by Larsson et al. were as high as 17% of interstitial transport in chromatography media [86]. Furthermore, use of superporous beads is associated with microfluidic devices that exhibit lower internal pressure [87]. Craters on the surface of the superporous bead lead to interconnecting cavities, forming long tunnels for easy access of fluids into the central region of the beads (Figure 10E,F).

The CRP capture in superporous beads is two times faster than homogenous agarose beads (Figure 10E) and for homogenous beads the assay time would need to be extended 5-fold in order to reach similar fluorescence levels from superporous beads. Higher penetration of analytes into the bead is afforded through control of agarose content in homogenous beads. Macropores in superporous beads are associated with increased access to otherwise inaccessible centrally located binding sites [63]. As shown in Figure 10F, superporous spheres evaluated by square penetration of signals exhibit analyte diffusivity that is 50× higher than internal transport within homogenous beads. These results suggested that the availability of large cavities enhances transport, thereby leading to shorter assay times.

Figure 11 shows the location of bound analytes with 3 different binding concentrations for anti-BSA capture antibody probes and at two different times of analyte delivery, using simulations. Unbound analytes’ access to the internal region of the bead is limited when capture probes are at high concentrations (Figure 11, bottom). The short persistence time of the unbound analyte results in high signal in the outer region. As capture sites at the outside of the bead become saturated, there is development of a moving boundary that penetrates to the center of the sphere. When the bead localized ligands become saturated, the unbound analyte molecules are then able to reach the internal capture probes. On the contrary, smaller capture probe amounts yield lower saturation intensities, but afford more rapid analyte binding in the bead interior (Figure 11, top). Here, the moving wave of bound analyte penetrates to the center of sensor ensemble significantly more quickly than for the case of higher loading levels. As shown in Figure 11, a uniform distribution of signal is developed for lower concentrations of capture antibodies, while a higher signal that is localized at the periphery of the bead is obtained for higher loading levels. Simulations, also validated by experimental studies, concur that, due to the fact the capture probes fall off at a fast rate, saturation of signal is quickly reached at both the bead exterior as well as internally within the spherical sensor element [75,76].

## 6. Image Analysis for the Bead-Based Programmable Bio-Nano-Chip (p-BNC) Sensor

As the above-described modelling studies reveal, the dynamics of biomarker capture are highly dependent on several key factors involving sample volume, biomarker concentration, antibody binding rates, bead localized capture antibody concentration, and flow rate. These factors contribute to in a predictable way to the spatial distribution of can be used to capture biomarkers. This location-dependent biomarker sequestration can be used to an advantage to further increase the performance of the bioassay. With this in mind, customized image analysis software was developed using novel computer vision methods. These automated routines allow for the identification of beads within the array and are used to analyze specified regions within the image (Figure 12) [88]. The method used first implements a Gabor annulus filter [89] to detect the bead location followed by the identification of an annulus region around the edge of the bead from which to extract mean pixel intensity. The entire analysis sequence is shown in Figure 12 and briefly described in its caption.

## 7. Evolution of the Smart Sensor Platform

For the last couple of decades, the fields of microfluidics and lab-on-a-chip have been plagued by significant problems related to the challenges of integration of all the lab components into the miniaturized chip structure. These efforts may be described as chip-in-the-lab rather than labs-on-a-chip. Given our goal to develop universal sensor tools with lab quality results that are scalable to global settings, there are key considerations in choice of instrumentation components, cost of reagents, and manufacturing methods that influence key decisions on the pathway to system’s level integration. Many of the elements of these choices fall outside of the scope of this review article, but relevant here is a general overview of the top-level description of the instrumentation architecture.

Figure 13 provides a summary of the main stages of instrumentation and chip sensor integration. For the early electronic taste chip sensor system, the imaging station was based initially on commercially available high-end fluorescent microscope that was equipped also for bright field measurements (Figure 13a). The microscope was fashioned with high end CCD-based imaging chips. A separate external fluid delivery system that was controlled by external computer was used to manage the fluid flow to silicon chips with flow-through bead micro-containers [88]. Commercially available software was used to control the image acquisition and the fluid flow. Manual data analysis was completed following the data acquisition steps. This early infrastructure was suitable for basic science activities as well as for the expansion of the menu of tests.

As the research focus evolved into more clinical themes, the requirement for ease of use for assay development was introduced and this led to the need for a higher level of coordination between the controlled fluid flow as well as the automated image acquisition. These efforts were supported by the p-BNC instrumentation shown in Figure 13b. In addition to creating a ‘work horse’ for bio-assay development, the structure also supported a dedicated effort devoted to increasing the level of integration and functions within the chip-based sensor suite. It should be emphasized that as external infrastructure of the lab is integrated into the chip, the miniaturized test ensemble needs to be adapted to accommodate the new features. This stage is where we devoted a significant amount of activity using a variety of rapid prototype methods.

Integrated multi-layer laminate labcards were built using xurography (i.e., razor blade writing), featuring fluid-routing channels, specimen metering, semi-permeable vent membranes, herring-bone mixers, buffer-filled blisters, built-in waste containers, and a UV-cured microchip for supporting the bead sensors. It should be noted that these labcards serve as an intermediate step that allows for the full functionality of the test structure to be demonstrated. These labcards contain the various components required to complete the fluid routing into the ‘business end’ of the sensor ensemble, that is, the microchip structure. The microchip structure serves as the key location at which the biomarker capture and signaling occur.

More recently, efforts have matured to the point where fully integrated instrumentation was developed and transitioned to a partner as required to reach a broad range of patients as shown in Figure 13c. At this point of the development cycle, sensor elements are produced by injection-molded plastic methods, and the resulting components are described as cartridges. Globally scalable instrumentation and cartridges are now being produced with high quality and under design control through a key commercial partnership (Section 8).

The various stages of sensor integration and the pathway to the universal sensor platform occurred through the following key steps:MACRO electronic taste chip (not integrated, chemistry of taste focus)MICRO p-BNC (partial integration, biomarker focus)
Bead configuration for solution phase detectionMembrane configurationMICRO platform to digitize biology (full integration, scalable cartridges)
Bead configuration cartridge for solution phase detectionMembrane cartridge configurationBoth configurations compatible with universal image instrumentation

## 8. Expansion of the Menu: Membrane-Based /Dual-Function Sensors and Additional Bead-Based Assays

Up until this juncture in the review, we have mainly described the bead-based miniaturized system for the identification and quantification of electrolytes, sugars, proteins, DNA oligonucleotides, therapeutic drugs, drugs of abuse, and toxins [64,65,90,91,92,93,94,95]. However, these beads had proven ineffective when challenged with cellular targets.

Therefore, in addition to exploring superporous beads, we also investigated adapting the miniaturized microfluidic systems with a membrane for the analysis of cellular samples. These membrane-based sensors were found to be an excellent tool for capturing and detecting particulate analytes, relevant to applications such as screening tools in the bioterrorism sector [96]), for the measurement of CD4 cell counts (a system with great utility potential for use in immune function monitoring of HIV-positive patients for resource-poor settings [97,98]), and in oral cancer diagnostics [99,100,101].

Furthermore, we have also described an integrated assay method suitable for the simultaneous measurement of CRP concentrations and leukocyte counts [102], both of which may serve as strong predictors for the development of coronary heart disease [103,104,105,106].

Collectively, two configurations of sensors both read through scalable imaging approaches (membrane-based for cellular analytes; bead-based sensors for soluble assays) (Figure 14) form a modular platform that exhibits a large menu of tests that are more typically serviced by an array of lab-based instruments. Both chip configurations are reprogrammed quickly as new information related to disease signatures is obtained from research settings. Please refer to Table 1 for a partial list of the test menu.

As mentioned above, the supported porous spheres afford shorter test times consistent with the goal of completing the biomarker panel measurements on the time frame of a doctor’s visit. The chip-based assay characteristics follow the Command Quals, that is, Cheap, Obvious, Miniaturized, Multiplexed, Automated, Nonperishable, Dependable, Quick, Unobtrusive, Adaptable, Limited (volume), and Self-contained [69,70].

The completion of high-performance multiplexed and multiclass assays remains a significant challenge for the pharmaceutical, healthcare, clinical, insurance, and biosensor sectors, emphasizing the critical need for a universal test system suitable for validation and implementation of biomarkers. High-impact diseases/addictions, including trauma, drugs of abuse, ovarian, oral, and prostate cancers, and cardiovascular disease have been the targets of our chip-based development and the focus of large-scale biomarker validation studies. Here, the capacity to use validated biomarker panels within a common diagnostic platform affords helping the bioscience community to move biomarker patterns into clinically actionable decisions, and it presents interesting synergies to access more effective wellness and treatment options.

The technology provides results in minutes, which makes them available at the POC, and, unlike most microfluidic approaches that employ planar arrays, the bead array sensors offer the unique feature of high surface area 3D beads to capture and most efficiently concentrate analytes.

Perhaps the most important feature of the sensor platform is its high-fidelity multiplexing capacity, i.e., detection of multiple analytes simultaneously. The technology’s chip-based assay platform allows for a high degree of multiplexing that is unique today, without compromising performance or sensitivity for low concentration analytes. Likewise, the p-BNC platform exhibits wide assay ranges, excellent limits of detection and quantitation, strong assay reproducibility, and most importantly a degree of multiplexing that allows for 8 or more biomarkers to be measured simultaneously. One sample draw and a single assay run on one sensor system can provide information on levels of multiple biomarkers in the sample and establish a disease risk score. More traditional approaches would require multiple blood draws and use a plurality of instruments. Table 1 provides a partial list of the biomarker diversity for the bead-based platform.

## 9. Sensor Integration into Clinical Flow

The laboratory version of the sensor bead-based methodology described above has been collectively involved in seven clinical studies, respectively, involving thousands of patients at numerous clinical sites for diseases/health conditions in the areas of cardiac disease [107,108], ovarian cancer [73,109], prostate cancer [100], and drugs of abuse [94,95]. These efforts are summarized in Table 2. Collectively, these clinical validation studies serve as a key step in moving this approach into broad clinical practice.

Figure 15 illustrates how the MICRO-form factor cartridge may be integrated into clinical flow. The first step of the process is the introduction of a small volume of patient specimen (∼100 μL or 2 drops of serum, plasma, or oral fluids) into the single-use cartridge. A portable analyzer accepts the cartridge and serves as the interface responsible for performing a sequence of steps for the assay, completed within the cartridge. Upon completion of the assay, image analysis routines convert bead signals into analyte concentrations which are delivered into statistical and machine learning algorithms for various clinical applications. The result is usually a single value for indicating a disease, for example, “Cardiac Score” for heart disease, or YES/NO for a qualitative test; results may then be depicted on a mobile health (mHealth) app. This technology has the potential to empower patients to be more active in their self-care through personalized health summaries.

Figure 16 illustrates the three major components of the MICRO-form factor technology: beads, cartridge, and analyzer. Figure 16A shows the MICRO-form factor’s bead technology from the vantage point of various length scales. Previously, bead sensor chips were anisotropically etched silicon 100 wafers. More recently, these early prototype labcards have transitioned to injection molding cartridges capable of being mass-produced and translated to eventual clinical use. The microfluidic cartridge contains embedded buffers and reagents enabling POC use. In most cases, a 4 × 5 array of flow-through microcontainers hosting an array of addressable agarose bead sensors is used. This agarose bead system offers the following advantages: (1) significantly reduced assay times as compared to diffusion-dependent ELISA, (2) customizable bead porosity enables detection of an array of physical and chemical properties, (3) an ability to execute two-site non-competitive and competitive immunoassays, and (4) ultra-low detection limits attributable to the relatively low nonspecific binding of the agarose three-dimensional structure. The bead configuration (Figure 16B) microfluidic cartridge streamlines various fluid handling steps rendering the technology as a true lab on a chip technology, in contrast to complicated lab-based interfaces that make the lab-on-a-chip devices unfeasible for use outside of the lab (i.e., “chips in a lab”).

To use the device, sample fluid is first introduced into the cartridge input port to fill the sample loop by capillary action, and a passive valve meters the sample to 100 µL. The cartridge device is closed using the adhesive cap after the sample is loaded and the sample fluid is processed by a series of filters before reaching the bead sensor matrix.

The cartridge also features two foil blister packs containing buffer (in most cases PBS). The blister packs are then ruptured by the instrument’s automatic fluid delivery module. The sample/antigen and detecting antibody delivery are controlled by the right and left blisters, respectively. Detecting antibody reagents stored in dried form within the cartridge are re-constituted with buffer and distributed to the sensors. Antigen and detecting reagents that remain unbound are washed away into waste fluid chamber on each side of the sensor matrix. Following a quick final wash with buffer, the assay is completed, and the cartridge is ejected and discarded. This integrated cartridge demonstrates strong utility for POC applications through a high degree of automation and integration, reducing handling errors and improving precision.

The analyzer weighs under 7 kg and contains a fluorescence imaging system, automatic blister actuation, internal computer, and touchscreen. The optical module images fluorescently labeled beads arrayed on a chip and via oblique LED illumination of the bead array. The imaging module uses an objective lens, emission filter, relay lens, and CCD camera affixed to a precision focusing stage.

To demonstrate the universal platform bioassay flexibility, a proof-of-concept experiment was completed on the integrated platform for prostate cancer, ovarian cancer, and heart attack (acute myocardial infarction (AMI)) screening and diagnostic applications (Figure 17).While the work described here demonstrates the capacity of the integrated bead array for flexible assay panels to support several disease indications, previous work using less integrated instrumentation provides more rigorous characterization for a menu of analytes and other relevant assay panels, including drugs of abuse, prostate cancer, ovarian cancer, and cardiac disease.

Another area of application of the bead-based sensor is in the area of allergy testing. Recently, a sandwich-type immunoassay was used for the measurement of total human (Figure 18) and allergen-specific IgE on the LOC platform (data not shown). The relevant immunocomplexes of the LOC-based assay for total IgE are shown in schematic form in Figure 18A. Briefly, a rabbit anti-human IgE antibody conjugated to the agarose bead is used to capture human IgE at the bead sensor ensemble. Detection of the captured analyte is achieved with a goat anti-human IgE antibody conjugated with a fluorophore. A typical result for this approach is shown in Figure 18B. Here, beads coated with antibody specific for human IgE produce a strong signal when exposed to a standard control containing increasing concentrations of human IgE, while negative control beads coated with an irrelevant to IgE antibody are not responsive.

A sandwich-type immunoassay can be used for the detection of allergen-specific human IgE system (data not shown). Here, allergen-coated agarose beads are used to capture human allergen-specific IgE antibody. Detection of the captured analyte is achieved with a goat anti-human IgE antibody conjugated with a fluorophore. The system detects total human IgE, as well as allergen-specific IgE (IgE for peanut, ragweed, cat, and dog epithelia and dust mite p1 and f). Furthermore, when the system is challenged with IgE-negative serum no detectable signal is observed. The method was positively correlated with those achieved with a commercial Enzyme Immuno Sorbent Assay (ELISA) kit.

Testing for drugs of abuse is also an interesting application for the bead-based approach [94,95]. When combined with noninvasive oral fluid sample collection methods, there is strong potential to improve the logistics of drug testing in both civilian and military application areas. Furthermore, this combination of a new drug testing platform and noninvasive sampling is ideal for onsite feedback for drug treatment centers. These application areas are now in development for this core technology. The approach utilizes a competitive type of immunoassay format as shown in Figure 19.

## 10. Artificial Intelligence-Linked Diagnostic Platforms for Early Cardiac Disease Detection

Artificial intelligence is experiencing explosive growth and interest in recent years as new AI applications are continuing to challenge, expand, and advance the capabilities of information technologies. There is enormous opportunity for AI systems in the healthcare industry with large volumes of untapped data resources in the forms of electronic health records, diagnostic test results, and clinical studies, to name a few. Despite the potential for improving patient outcomes and reducing the healthcare costs, the adoption of AI applications in health care has lagged other industries due to significant barriers. A recent survey of over 50 executives of healthcare companies using AI has identified a few of these barriers, including: (1) the lack of case studies to convince stakeholders about the return on investments in AI applications; (2) the lack of interpretability in AI models (i.e., the “black box” problem); and (3) the complexity of stakeholder relationships in health care, from the hospital executives, to the professionals using the technology, and ultimately to the explanation of benefits to the patients [111]. Despite their deep potential to impact health care, new technologies that utilize AI and predictive analytics in medical diagnostics and lab-testing applications are additionally hindered by regulatory challenges and data privacy concerns. Laboratory testing plays a vital role in clinical decision making. However, today we lack a standard point of care tool for both special- and general-purpose clinical lab testing. This growing list of heterogeneous point-of-care tools complicates the issue of patient management and data flow.

The McDevitt group is attempting address these challenges directly and bridge the gap between predictive analytics and a ‘universal platform to digitize biology’. While there are countless biomedical applications for these ‘sensors that learn’, perhaps the area in which this technology may have the greatest social and economic impact is in chronic diseases, such as cardiac heart disease—the leading cause of death globally. This section is a case study for the McDevitt group’s efforts towards applying artificial intelligence to bead array measurement results and traditional risk factors for assessing a spectrum of cardiovascular health/disease indications.

The Cardiac ScoreCard is a clinical decision support system (CDSS) in development by the McDevitt group that uses LASSO logistic regression to transform multiple risk factors and biomarker measurements into a one score with intuitive and clinically relevant information. Once completed, the Cardiac ScoreCard can provide personalized reports for a range of CVDs with diagnostic and prognostic models for cardiac wellness, AMI, and heart failure (HF).

The ScoreCard assay currently hosts a multiplexed array of markers selected for their involvement in various stages of the cardiac disease cascade, including those clinically silent stages of the atherosclerotic plaque development, thereby providing coverage of a diverse patho-cardio-physiology (Figure 20) [108]. Through this broad biomarker coverage, the Cardiac ScoreCard technology can model a range of clinical and subclinical sequelae of atherosclerosis, and preliminary models for cardiac wellness and HF diagnosis have been developed with this approach.

A cardiac wellness model was developed using a risk-stratification approach to dichotomize subjects into low- and high-risk groups based on the presence of one or more risk factors, including history of stroke, coronary artery disease, diabetes, myocardial infarction, hypertension, and high cholesterol. This cardiac wellness ScoreCard model demonstrated improved discrimination performance over both the Framingham Risk Score (10-year CVD) and a model comprised only of biomarker predictors (area under the curve (AUC) of 0.84, 0.79, and 0.77, respectively) (Figure 21). Similarly, a diagnostic model for HF was developed and compared to the results of a BNP test. The multi-biomarker HF diagnosis ScoreCard demonstrated improved discrimination performance over the single-marker BNP test (AUC of 0.94 and 0.93, respectively). These results suggest that the consolidation of information-rich biomarkers and risk factors into statistical learning algorithms may enhance the diagnosis of CVDs and the quantitation of overall cardio-wellness.

## 11. Point of Care Oral Cytopathology Tool (POCOCT) for Precision Oncology

Oral cancer is a worldwide health problem afflicting ~300,000 people each year. In the US, approximately 50,000 oral and pharyngeal cancers (OPC) are diagnosed annually (10/100,000 incidence). High mortality associated with OSCC is often attributed to the advanced disease stage of many OSCCs upon initial identification and surgical biopsy. The overall 5-year survival rate is ~64%, yet when diagnosed at an early stage (i.e., stage I/II), the rate increases dramatically to ~83%. These OSCCs, estimated to comprise more than a third of oral and pharyngeal cancers, are usually preceded by potentially malignant oral lesions (PMOLs), underscoring the need for new diagnostic methods targeting early tumor progression and malignant transformations.

The membrane-equipped platform allows for cytological analysis from samples collected via a minimally invasive brush biopsy method. This automated cytology platform simultaneously performs cell morphometry and molecular biomarker expression measurements. Subsequent image analysis and statistical learning techniques produce a numerical index capable of classifying a range of clinically relevant oral lesion determinations, turning around biopsy results in a matter of minutes rather than days for conventional pathology and allowing the technology to perform cytology measurements at the POC. The POC testing is expected to have tremendous implications in the management of patient disease by enabling dental practitioners and primary care physicians to circumvent the need for multiple referrals and consultations before obtaining an assessment of the molecular risk of PMOLs.

We have developed robust classification models based on high content analysis at the single cell level (Figure 22, Figure 23 and Figure 24) [112,113,114,115]. More specifically, samples from 714 prospectively recruited subjects with suspicious oral lesions across six diagnostic categories were tested using our ‘cytology-on-a-chip’ and confirmed by scalpel biopsy and histopathology. More than 200 cellular features, including biomarker expression and nuclear/cellular morphology, were measured for every individual cell with an average of 2000 cells imaged per patient. Dichotomous models were developed for various lesion determinations, including a binary “low-risk/high-risk” model, which resulted in an AUC of 0.84. The model accuracy demonstrated an improvement over the preliminary agreement rate of a pool of expert pathologists (70% vs. 69%, respectively). Important predictors defined by the models were nuclear-cytoplasm (NC) ratio, nuclear and cell area, cell circularity, and Ki67 and EGFR expression (Figure 23 and Figure 24).

This cytology-on-a-chip approach may provide a useful tool for monitoring patients with suspicious lesions and serve as a platform for discovering new insights in the early stages of OSCC.

## 12. Discussion

Despite large investments in translational research, the majority of life sciences research programs remain largely decoupled from real-world clinical practice and continue to struggle translating their inventions to broad clinical use. Typically, the device development process, whether in academia, national labs or the industrial sector, involves a sequence of time-consuming steps that, more often than not, happen in a linear, less efficient and disjointed manner, draining precious venture capital resources and federal funding, alike. Prior to pursuing clearance of a new device, product development teams need to assemble substantial evidence in support of the candidate device for its intended use. Too often today, academic results are decoupled from the detection method in the end-user setting. Upon selection of biomarker and detection modalities, efforts are completed to develop the candidate product prototype. This juncture serves as the major ‘pinch point’ for diagnostic device translational efforts.

A striking analysis by Schully and coworkers of the extramural grant portfolio of the National Cancer Institute (NCI) from Fiscal Year 2007 explains exactly what the reasons are for the dismal rate of translation of new medical tests into real-world practice [116]. The group classified both funded grants and publications with respect to the stages of bench-to-bedside translation as follows: T1—Research on the development of candidate health device or therapy application; T2—Evaluation of a candidate application and development of recommendations based on evidence; T3—Research that assesses how to integrate an evidence-based recommendation into cancer care and prevention; and T4—Assessment of health outcomes and population impact.

Importantly, the Schully team found that only 1.8% of the grant portfolio and 0.6% of the published literature reached the T2 stage or beyond. Furthermore, in the precision diagnostics area the translation gap appears even larger. Overall, these findings suggest that it is necessary to develop new bridges that can span the disparate disciplines that have been historically isolated and that have led in part to these translation gaps.

Likewise, the McDevitt group is in the process of establishing the “Early Disease Detection through Distributed Diagnostic Devices” (E_5_D) program (Figure 25). In collaboration with our corporate partner, SensoDx, a globally competitive infrastructure will be established that bridges the scientific, engineering, clinical, economic, product realization, and regulatory aspects of smart precision diagnostic device commercialization. It is this transition zone (i.e., the steps toward initial qualification and usage models for eventual clinical use) that serves as the focus for the E_5_D program. This program will bring together shared resources including carefully curated clinical samples, operating procedures, electronic data acquisition systems, and biostatistics. A key step to completing this process is the development of a pathway to move past the publication and into a situation where these programmable sensors can be used by clinicians and the bioscience community for development of the next generation of diagnostic panels suitable for management and treatment of patients. The universal instrumentation being developed now by SensoDx has the potential to empower a wide variety of disciples to develop the next generation diagnostic panels for health and wellness applications (See Figure 26).

## 13. Conclusions

This article has described the evolution of chip-based sensor systems with an initial focus on an electronic taste chip system. The original concept focused on the development of smart sensor systems that could combine a plurality of chemistries en route to making quality decisions about complex fluid analysis. With time, the focus moved into clinical themes where biomarker patterns tied to patient health status could be monitored. The efforts were steered in part by the pull of this diagnostic technology to service an ever-expanding list of soluble biomarkers that are being discovered through the omics disciplines. The linkage to cell-based testing using a universal image-based platform has also been completed. The ability to complete single cell multiparameter oncology applications at the point of care for the first time opens up a range of new exciting clinical themes for pharmaceutical research as well as for the precision oncology space.

Most of the prior biomarker-based efforts remain largely decoupled from the practice of medicine, in part due the significant challenges associated with moving biomarker panels through the rigorous process of discovery, validation, and clinical implementation. The tools developed here have strong potential to establish a ‘biomarker highway’ that provides a vehicle to accelerate the translation of biomarker panels into clinically actionable tools.

For the areas of oral cancer and cardiac heart disease, these tools have been used to acquire first in kind data bases where new insights into disease specific phenomena can be observed. The linkage to large data sets and the use of machine learning to extract patient specific information with potential to accelerate the arrival of the precision medicine area is of strong interest.

## Figures and Tables

**Figure 1 micromachines-10-00251-f001:**
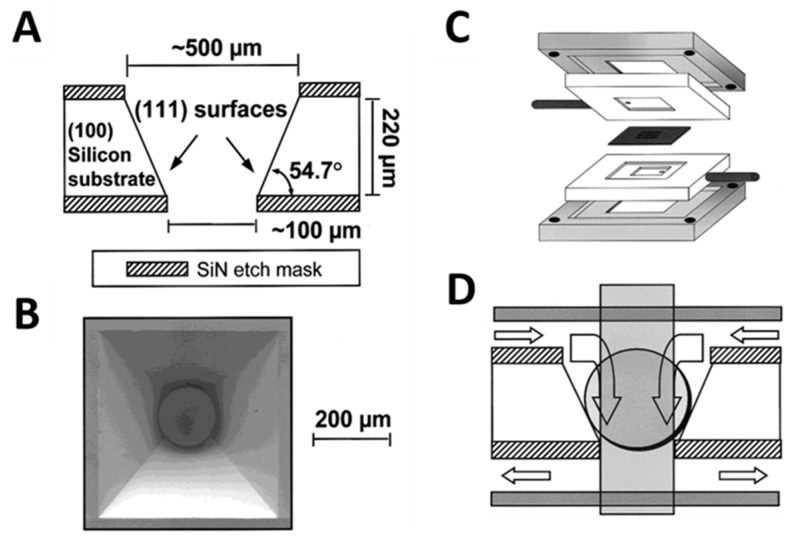
The illustration shows various elements that are related to the electronic taste chip. In panel (**A**), a diagram is provided displaying the pit region within the silicon chip where bead sensors are localized. In panel (**B**), a top perspective of a single well wherein a resin sensor bead is located centrally is shown. In panel (**C**), a schematic diagram depicting one of the flow cell interfaces developed by the McDevitt group is included. In panel (**D**), a schematic is shown of the bead confinement strategy, fluid direction (arrows), and optical access (light gray shaded area) within the flow cell. Adapted from [66] with permission.

**Figure 2 micromachines-10-00251-f002:**
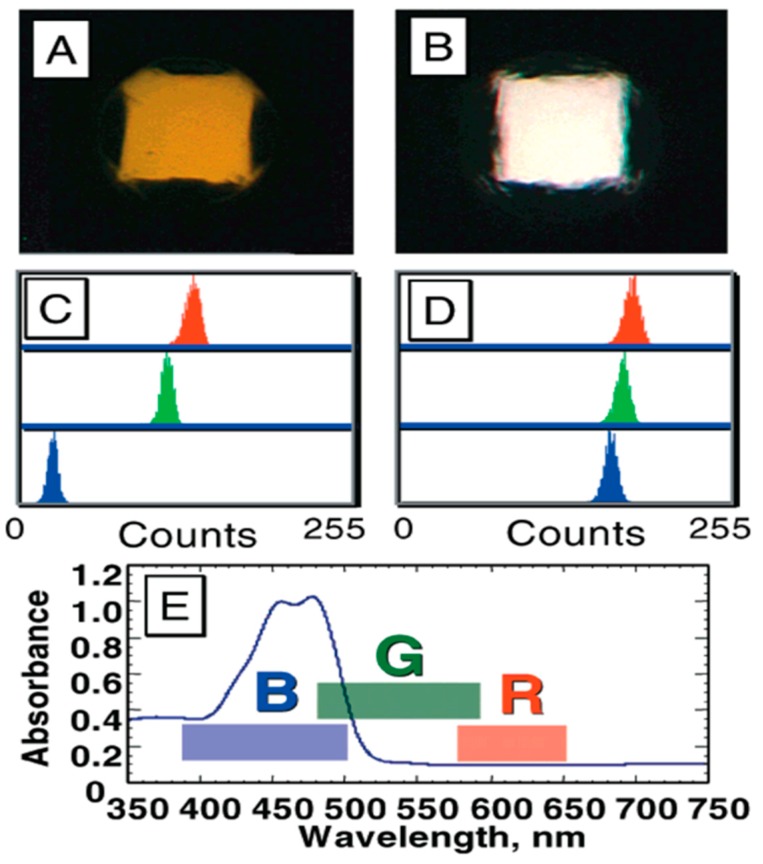
Optical intensity information from the video chip recorded simultaneously at different bead sites were acquired by the electronic taste chip platform. The bead derivatized with fluorescein appears yellow (**A**), while the reference/blank microsphere shows white (**B**). The effective color attenuation is shown for (**C**) fluorescein derivatized microsphere and (**D**) the blank microsphere. This data is used to quantitated RGB color histograms observed in each of these spectral regions. The bottom segment (**E**) provides the approximate RGB spectral features and the absorbance characteristics of fluorescein. Adapted with permission from [66].

**Figure 3 micromachines-10-00251-f003:**
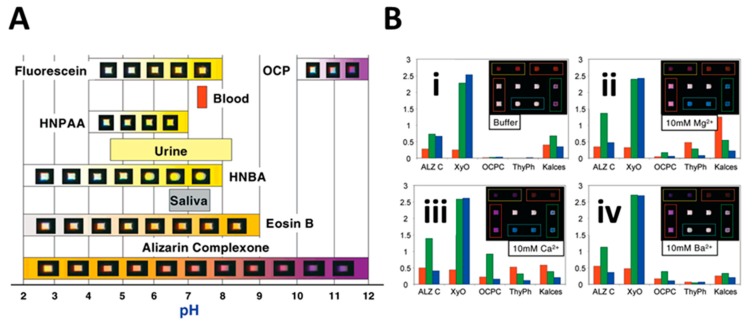
This illustration provides a summary of data for the applications of the electronic taste chip. (**A**) For sour detection, a series of optical responses for a total of 6 separate pH indicator dye-coated microspheres. (**B**) For the electrolyte-salt group, green, blue, and red responses for a series of complexometric dyes, as observed for 4 different solutions: (i) pH 9.8 inorganic buffer; (ii) pH 9.8 buffer with 10 mM MgCl_2_; (iii) pH 9.8 with 10 mM Ca(NO_3_)_2_; (iv) pH 9.8 containing 10 mM BaCl_2_. Insets show optical micrographs of the beads. Adapted from [66] with permission.

**Figure 4 micromachines-10-00251-f004:**
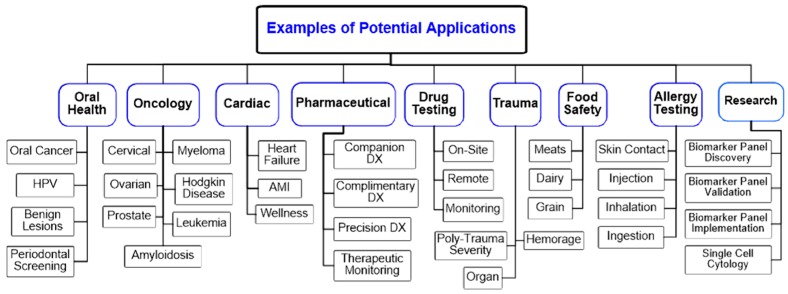
A broad range of applications have been demonstrated for the p-BNC sensor approach. The various test sectors are summarized in this illustration.

**Figure 5 micromachines-10-00251-f005:**
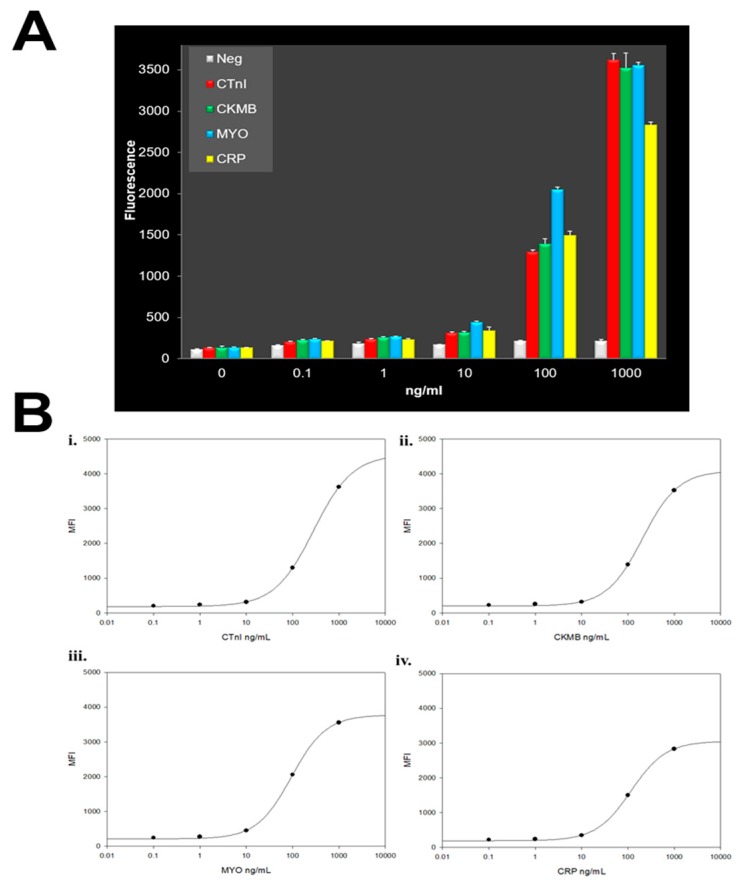
This illustration shows dose response curves for cardiac assays as developed on a partially integrated p-BNC sensing platform. These data were acquired with syringe pump fluid control using a laminate labcard. The measurements of the four shown cardiac biomarkers are completed within a single multiplexed assay. The sample matrix used here is serum.

**Figure 6 micromachines-10-00251-f006:**
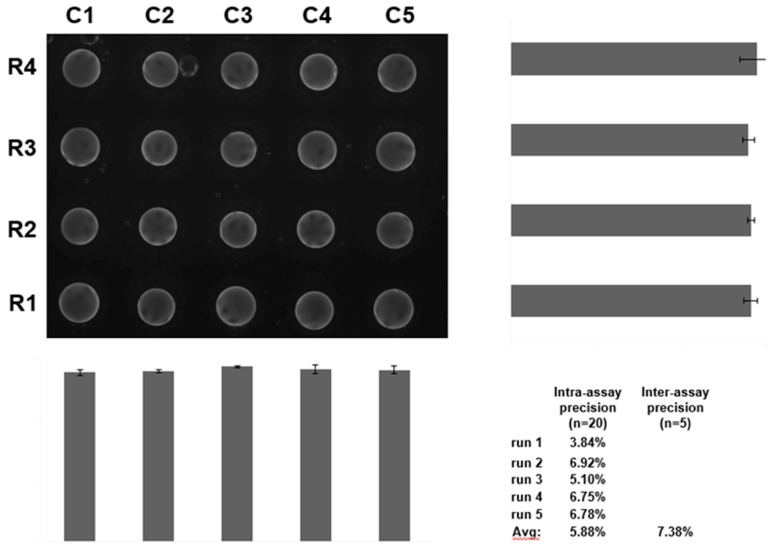
Precision data acquired for the cardiac biomarker myoglobin on a partially integrated p-BNC platform (see Figure 5 for more details).

**Figure 7 micromachines-10-00251-f007:**
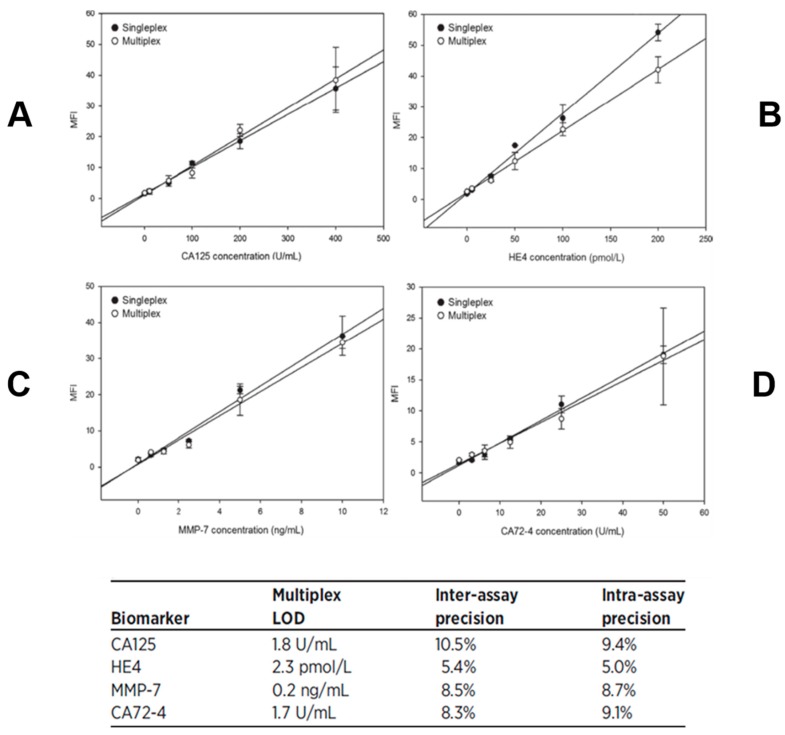
Calibration curves are shown for the following cases: (**A**) CA125, (**B**) HE4, (**C**) MMP-7, and (**D**) CA72-4. The data was obtained for both singleplex (black circles) and multiplex (white circles) formats. The insert table at the bottom lists the assay characteristics of the multiplexed test, such as limits of detection (LOD) and intra- and inter-assay precision values for same analytes. Adapted from [73] with permission.

**Figure 8 micromachines-10-00251-f008:**
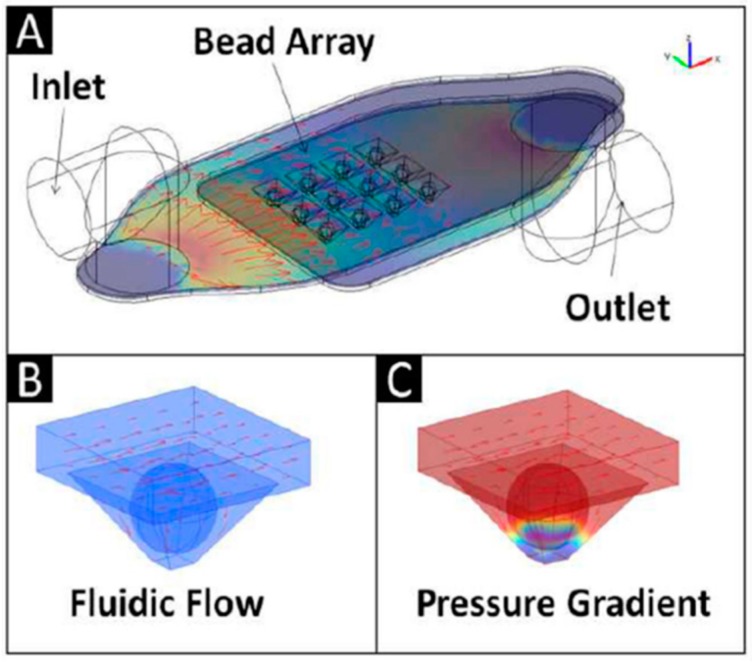
This illustration provides three-dimensional computational modeling for fluid delivery within agarose bead ensembles. Show here are the following cases: (**A**) Schematic showing fluid delivered via an inlet channel to a porous bead array; (**B**) A high magnification perspective of a single bead revealing that signal develops at the exterior of the bead. (**C**) Pressure is found to influence the flow rates at the bead well interface. Adapted from [74] with permission.

**Figure 9 micromachines-10-00251-f009:**
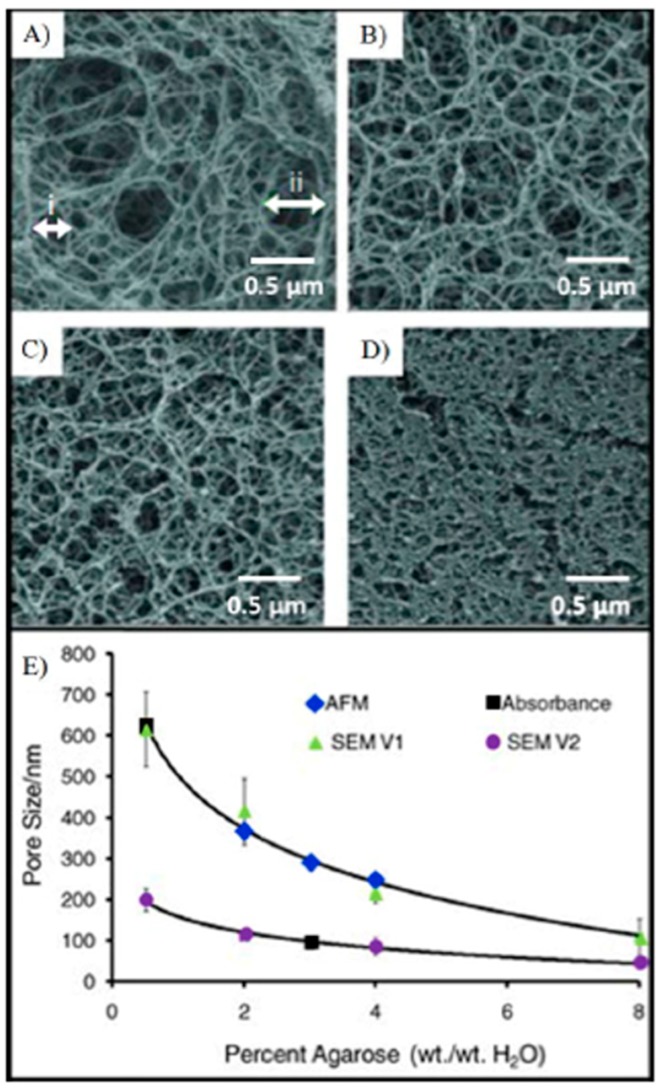
The illustration shows the influence of weight fraction of agarose content controls range of pore size of the beads. Here, electron microscopy images of interior of beads of varying agarose concentrations are provided as follows: (**A**) 0.5% (wt/wt%); (**B**) 2% (wt/wt%); (**C**) 4% (wt/wt%); and (**D**) 8% (wt/wt%). The various cases reveal that decrease in pore size as the amount of agarose increases. Panel (**E**) shows inverse relationship for pore size and agarose percentage, as extracted by microscopy techniques. Adapted from [75] with permission.

**Figure 10 micromachines-10-00251-f010:**
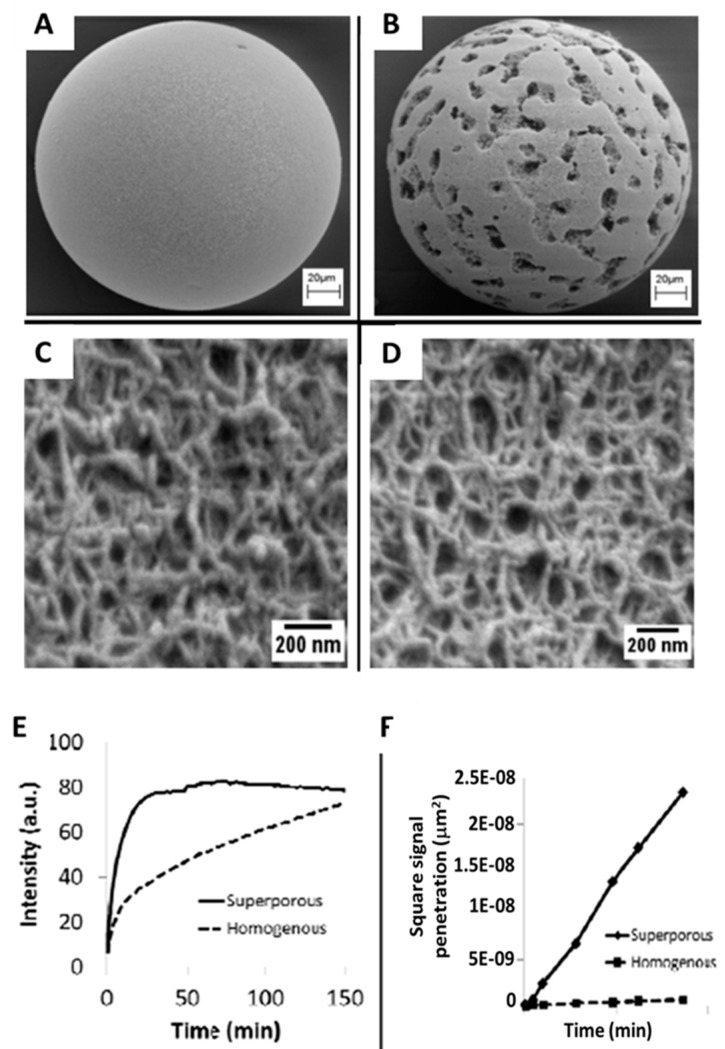
The illustration shows the comparison between homogeneous and superporous beads for the following cases as obtained by SEM for panels A to D. (**A**) Images reveals surface morphology of homogenous 4% agarose beads; (**B**) 4% superporous beads with ~30 µm microcavities; (**C**) higher magnification of homogeneous case; (**D**) superporous bead also at detailed view. The bottom two panels show: (**E**) Comparison of CRP capture by homogeneous and superporous agarose spheres; (**F**) the CRP diffusion is 50x higher in superporous beads. Adapted with permission from [75].

**Figure 11 micromachines-10-00251-f011:**
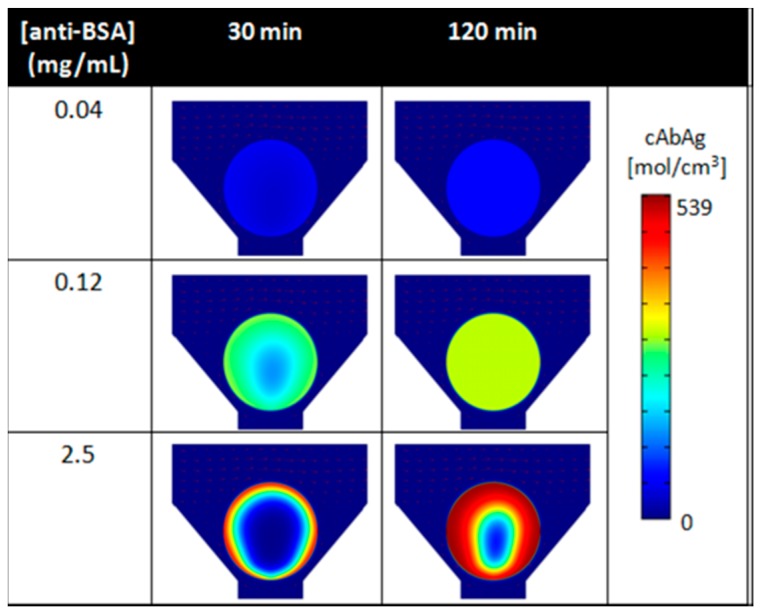
The illustration shows the elucidation of the spatial distribution of fluorescently labeled BSA via finite element analysis as a function of time and microbead receptor concentration. Adapted from [75] with permission.

**Figure 12 micromachines-10-00251-f012:**
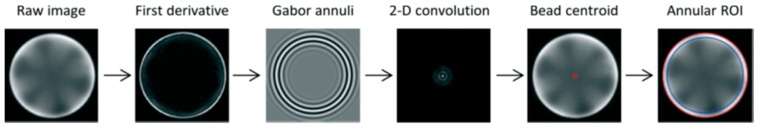
Summary of the bead analysis algorithm shown for one bead in the array. The first-order difference of the raw image is calculated to elucidate the edge of the bead. A series of Gabor annulus filters are then convolved over the first difference image. The 2-D convolution response is then used to approximate the center of the bead. The bead’s outer edge is identified by the maximum gradient in pixel intensity. Since the bead signal develops nonuniformly, an annulus region of interest is mapped in which the region of highest intensity along the outer diameter is extracted and averaged. Adapted from [88] with permission.

**Figure 13 micromachines-10-00251-f013:**
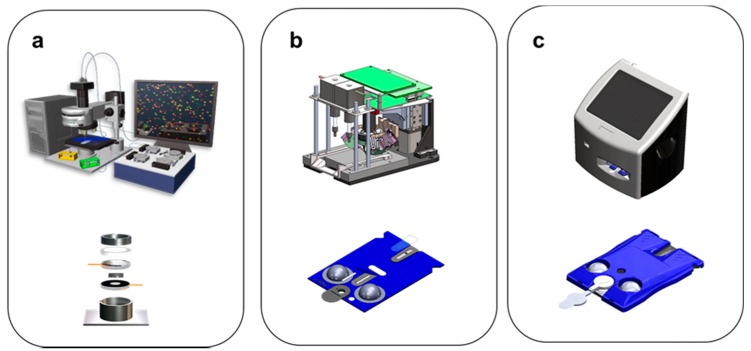
The evolution of the instrumentation is shown. (**a**) First, a typical MACRO-lab-based- configuration used in the early stages. This structure has a microchip element that is supported by external pumps, valves, and waste chambers. (**b**) Second, a typical MICRO non-form factor prototype configuration that serves as an intermediate step with partial integration is shown. This structure has a labcard for fluid routing and supports a microchip sensor where biomarker capture occurs. (**c**) The final MICRO-form factor prototype configuration serves as key step towards full integration both in terms of the instrument and the injection molded plastic cartridge. The universal cartridge comes in both bead and membrane configurations that are serviced by image-based instrumentation.

**Figure 14 micromachines-10-00251-f014:**
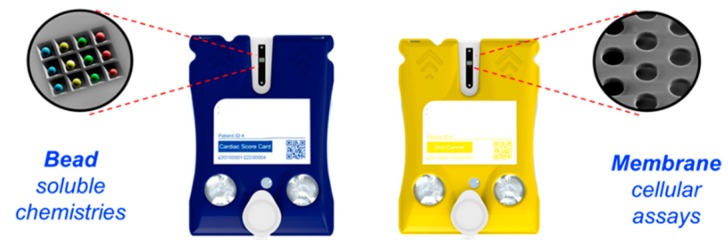
Bead- and membrane-based assay platforms of the MICRO-form factor cartridges dedicated to soluble chemistries and cellular assays, respectively.

**Figure 15 micromachines-10-00251-f015:**
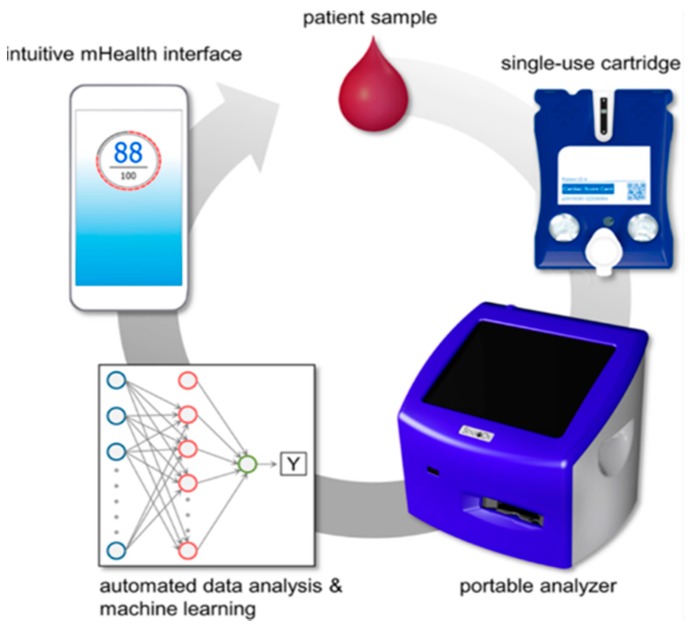
Intended cycle of use of the MICRO-form factor instrument and cartridge. Adapted from [110] with permission.

**Figure 16 micromachines-10-00251-f016:**
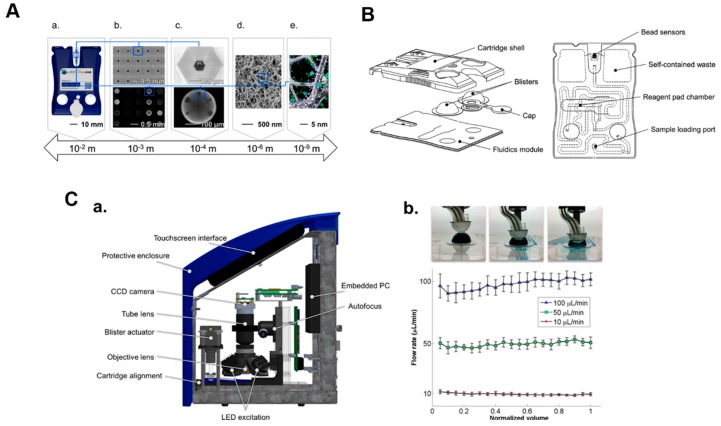
The MICRO-form factor’s system components. (**A**) Features of the sensor technology at multiple length scales: (a) MICRO-form factor cartridge; (b) matrix with (bottom) and without (top) bead sensors; (c) an agarose bead (bottom) and microcontainer (top); (d) SEM image showing porous agarose bead structure; and (e), illustration of an agarose bead fiber functionalized with immunoreagents. (**B**) Illustration of cartridge features. (**C**) Summary of MICRO-form factor instrumentation: (a) illustration of the prototype analyzer; (b) fluid delivery system with images of an actuator compressing a blister (top) and flow rate verification results (bottom) for five runs at low, medium, and high target flow rates. Adapted from [110] with permission.

**Figure 17 micromachines-10-00251-f017:**
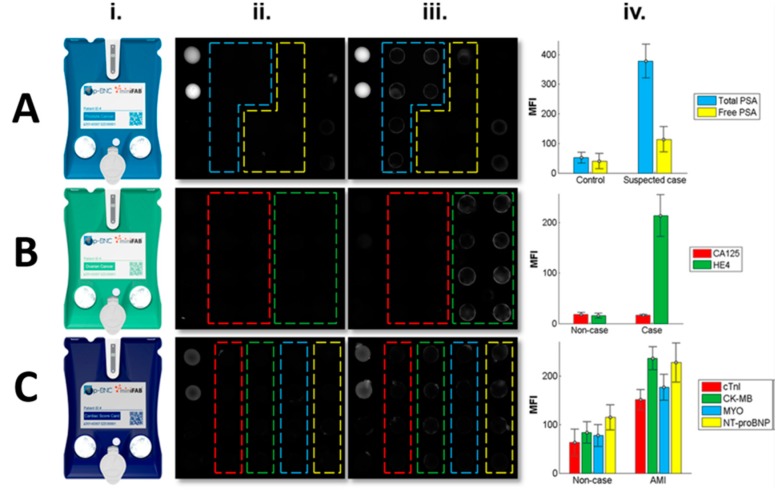
Demonstration of bead-based cartridge for multiplexed panels for prostate (**A**) and ovarian (**B**) cancers and AMI (**C**) using consumable cartridges (i). Bead sensor images for non-case/control (ii) and case (iii). The prostate cancer panel includes total and free prostate-specific antigen (a). The ovarian cancer panel includes CA125 and HE4 (b). The AMI diagnostic panel includes cTnI, CK-MB, myoglobin and NT-proBNP (c). Average fluorescence intensity measurements are shown for each biomarker (iv). Reprinted with permission from [110].

**Figure 18 micromachines-10-00251-f018:**
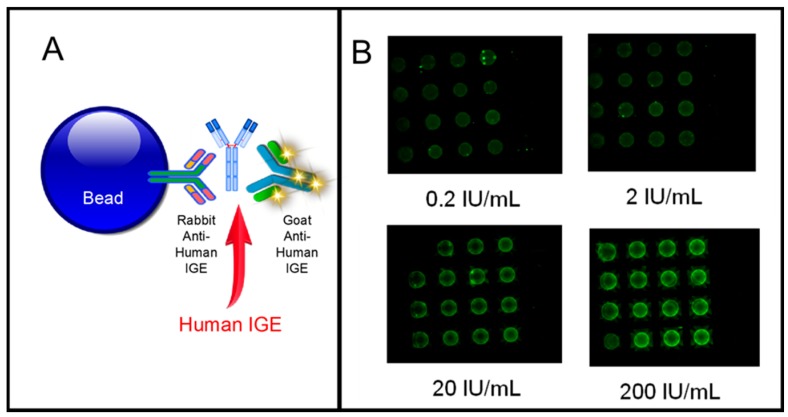
(**A**) Allergy testing on bead-based assay platform showing total human IgE assay immunoschematic; (**B**) Images obtained using MACRO non form factor instrumentation of bead array exposed to increasing concentration of human IgE. The platform has also been used to identify allergen-specific IgE as well (data not shown).

**Figure 19 micromachines-10-00251-f019:**
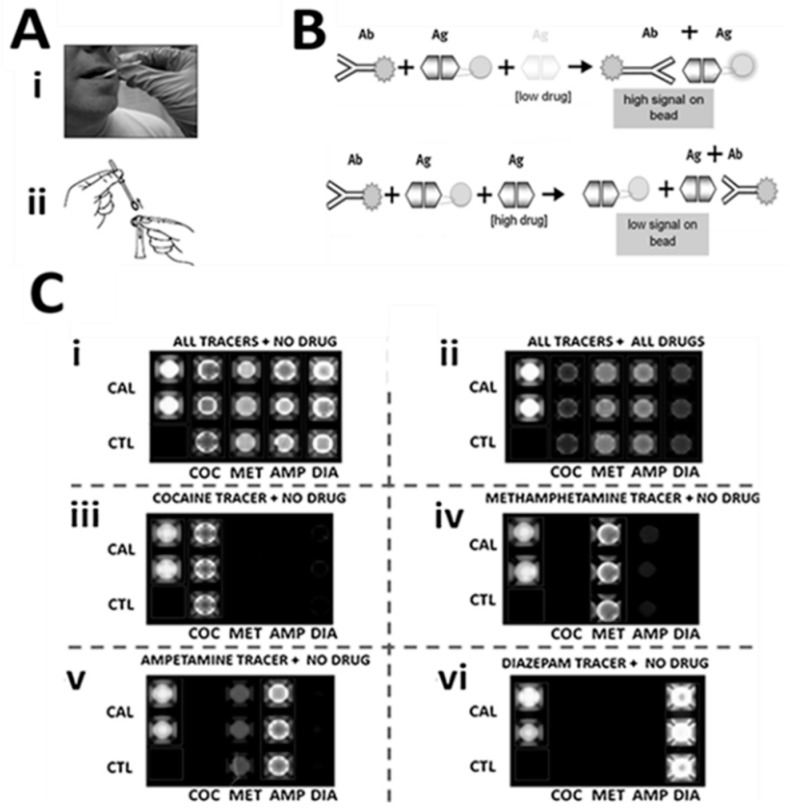
Saliva-based tests for drugs of abuse on the bead-based sensor suite. (**A**) Oral fluid sample collection by a swab (A-i) and extraction (A-ii). (**B**) Immuno-components of this competitive assay approach. (**C**) Series of experimental runs demonstrating the specificity of this assay system. Adapted from [94] with permission.

**Figure 20 micromachines-10-00251-f020:**
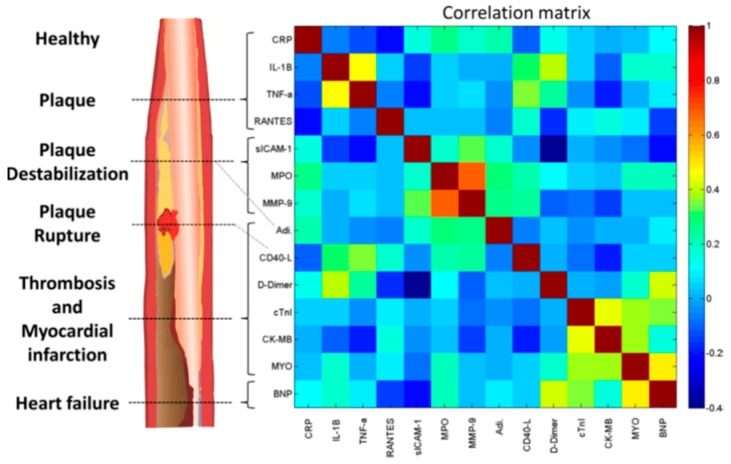
The CVD pathophysiology and Pearson correlation coefficient matrix-based grouping of cardiac biomarkers. The illustration on the left outlines atherosclerotic plaque stages and associated biomarkers. On the right is shown the Pearson correlation coefficient matrix calculated from serum biomarker concentrations of 579 patients in the study. Adapted from [108] with permission.

**Figure 21 micromachines-10-00251-f021:**
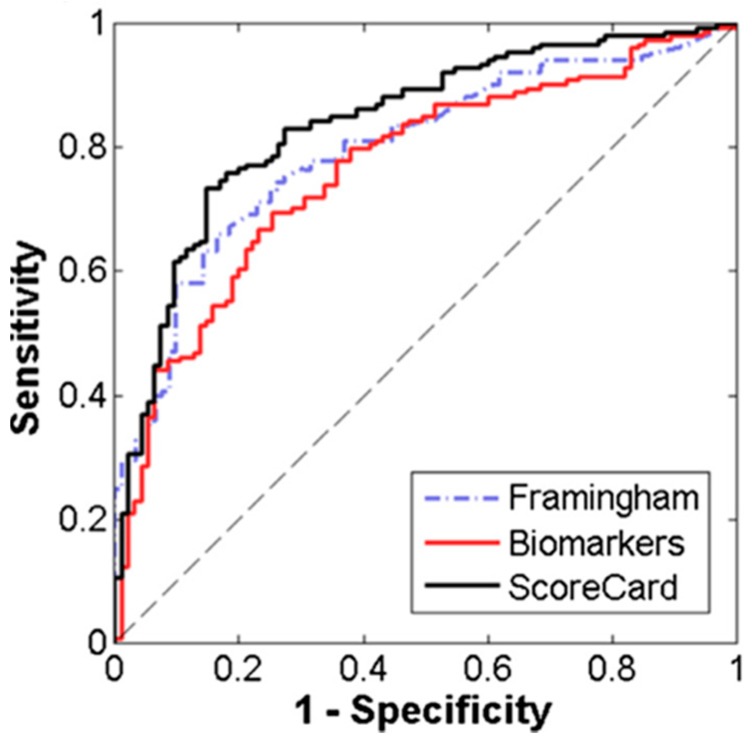
Receiver Operating Characteristic curves show improved discrimination performance for the cardiac wellness ScoreCard model over Framingham Risk Score and a biomarkers-only model. Adapted from [108] with permission.

**Figure 22 micromachines-10-00251-f022:**
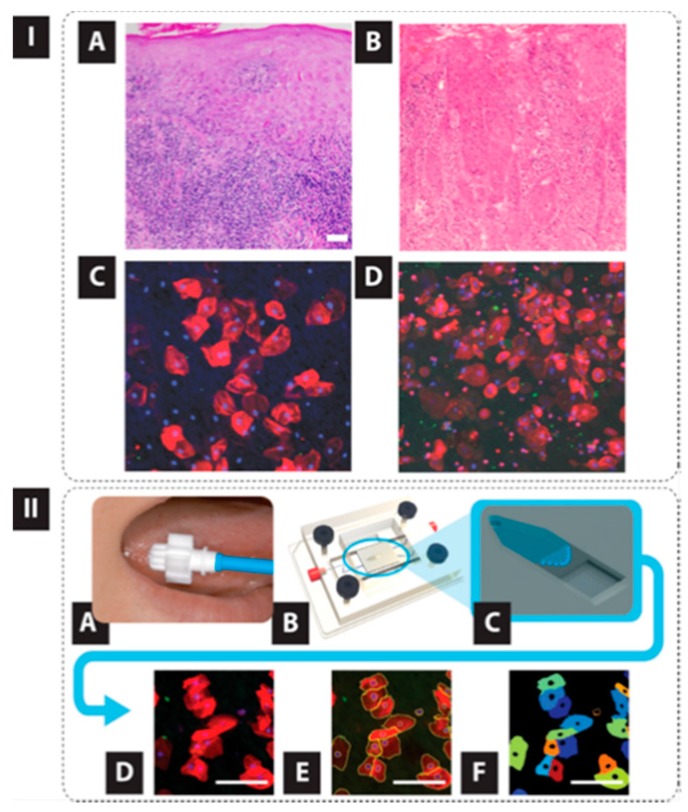
Cytology-on-a-chip. Panel (**I**) the “cytology-on-chip” approach vs histopathology. Histopathological (H&E staining) images (A,B) and immunofluorescence cytology images (C,D) for four different patients. Panels A and C are derived from healthy controls and panels B and D from OSCC patients. Panel (**II**) “Cytology-on-chip” test in which a brush biopsy sample is obtained (A), loaded into the flow cell (B) which captures the cells on a porous membrane (C). Multispectral fluorescence images are captured (D), automated image analyses identify individual cells (E), and important intensity and morphology measurements are extracted to be used in the machine learning algorithms (F). Reprinted from [115] with permission.

**Figure 23 micromachines-10-00251-f023:**
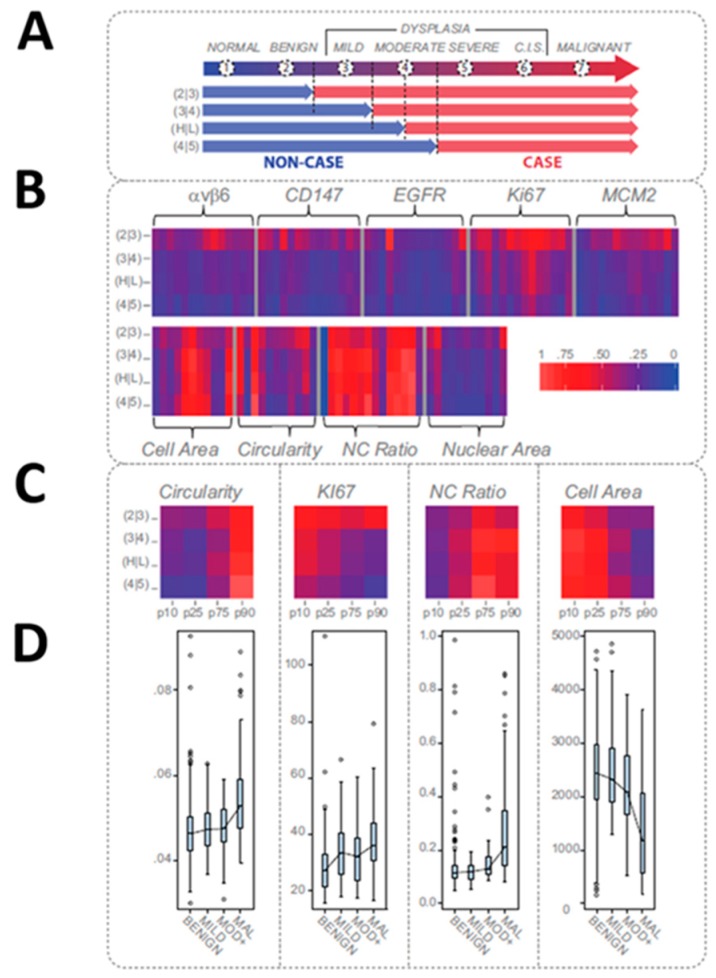
Summary of results from Random Forest model development. (**A**) The OSCC diagnostic spectrum and four binary splits used to dichotomize outcomes into ‘‘Case” or ‘‘Non-case”. (**B**) Normalized Gini value heat maps from Random Forest models demonstrating variable importance of select predictors across four diagnostic splits (y axis). (**C**) A subset of predictors from (**B**) showing summary percentile measurements (10, 25, 75, 90 percentiles). (**D**) Box plots displaying the median value distributions for circularity, Ki67, NC ratio, and cell area. Adapted from [115] with permission.

**Figure 24 micromachines-10-00251-f024:**
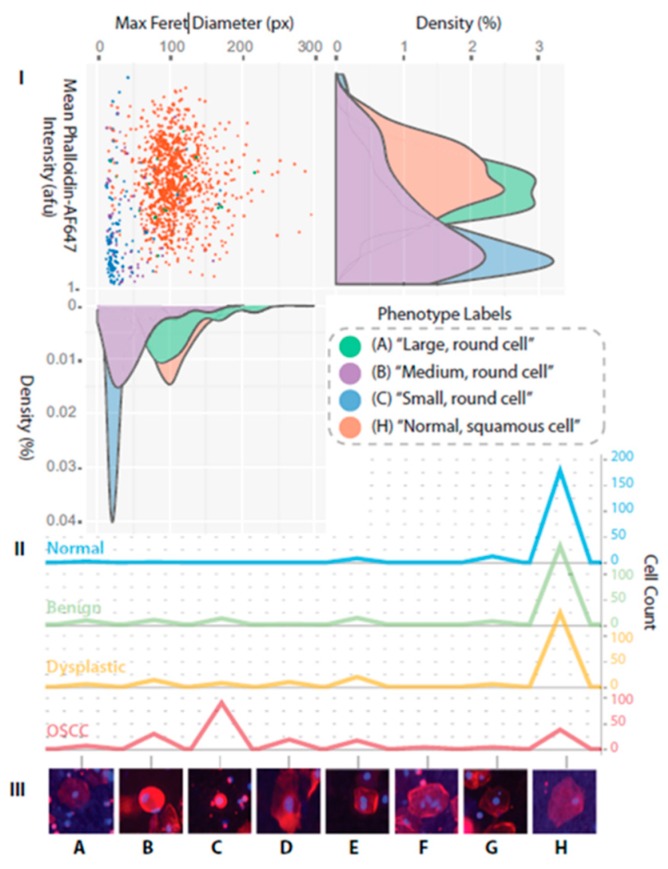
Cell phenotypes classified by morphometry. Panel (**I**) scatterplot and histograms for maximum Feret diameter and mean fluorescence intensity for Phalloidin. Panel (**II**) cell counts for each phenotype shown in Panel (**III**) for 300 cells selected at random from subjects with lesion determinations of Normal, Benign, Dysplastic, and OSCC. Panel (**III**) representative images of cell phenotypes with unique morphological characteristics, including: (A) cells with high circularity, low NC ratio; (B) cells with high circularity, high NC ratio, and medium cytoplasm area; (C) cells with high circularity, high NC ratio, and small cytoplasm area; (D) large cells with enlarged nuclei; (E) binucleated cells; (F) polynucleated cells; (G) cells with micronuclei; and (H) normal squamous cells. Reprinted from [115] with permission.

**Figure 25 micromachines-10-00251-f025:**
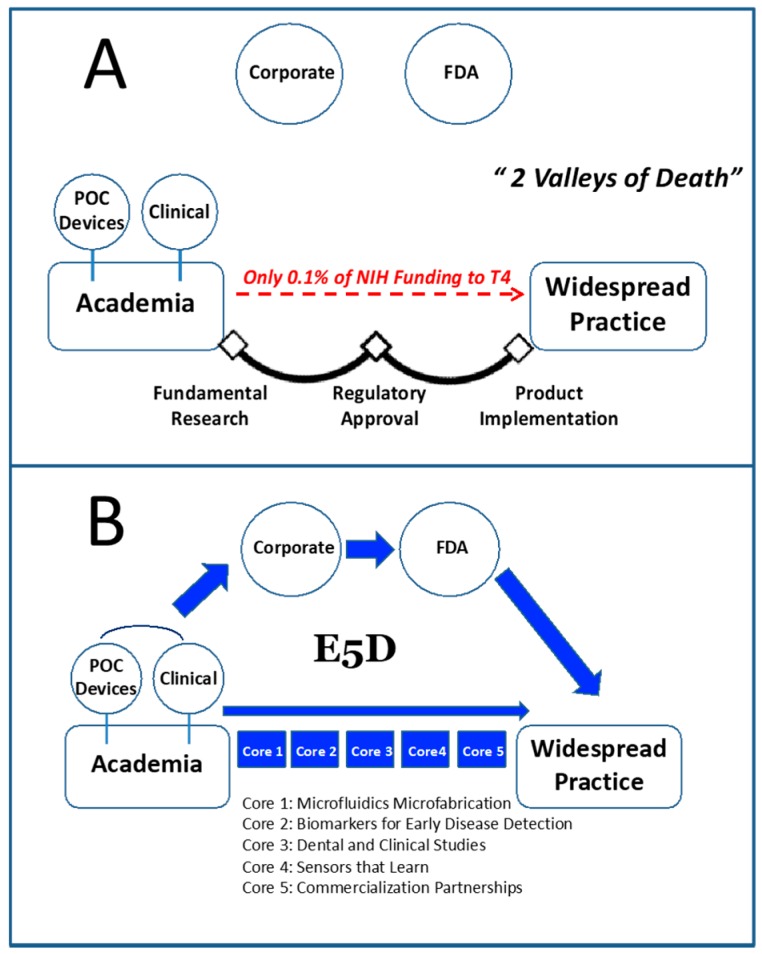
(**A**) The two valleys of “death” that medical device manufactures face and (**B**) the E_5_D bridge that connects academic findings with their widespread usage at the clinical setting (to stage T4 and beyond).

**Figure 26 micromachines-10-00251-f026:**
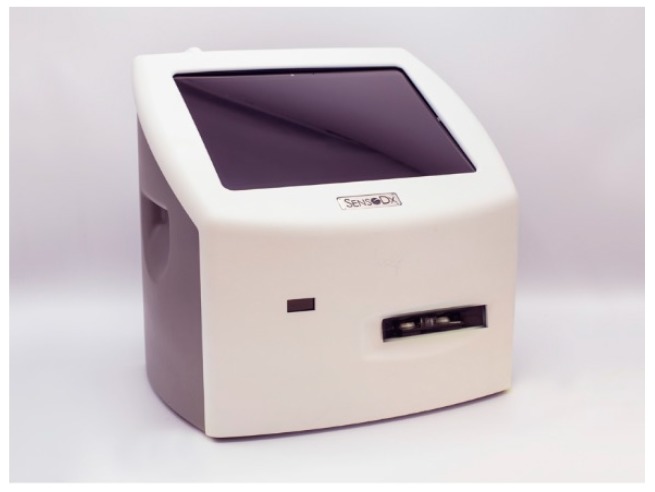
Picture of the current SensoDx instrumentation. The instrument is fully integrated with cloud connected capabilities and able to complete complex tests using single push of button process.

**Table 1 micromachines-10-00251-t001:** Partial list for bead-based assays. Adapted from [75] with permission.

Biomarker	Clinical Use
C-reactive protein	AMI, Risk Definition
Soluble CD40 ligand	Cardiac Risk Definition
Monocyte chemoattractant protein-1	Cardiac Risk Definition
Myeloperoxidase	Cardiac Risk Definition
Myeloperoxidase (multiplexed)	Cardiac Risk Definition
Interleukin-1beta	Cardiac Risk Definition
Interleukin-6	Cardiac Risk Definition
Tumor necrosis factor-alpha	Cardiac Risk Definition
Cardiac troponin I	AMI Diagnosis
Myoglobin	AMI Diagnosis
CK-MB	AMI Diagnosis
Apolipoprotein A1	Risk for AMI recurrence/Prognosis
Apolipoprotein B	Risk for AMI recurrence/Prognosis
Brain natriuretic peptide	Congestive Heart Failure
N-Terminal proBNP	Congestive Heart Failure
Human serum albumin	Cardiac Risk Assessment
Transferrin	Blood contamination in saliva
Carcinoembryonic antigen	Ovarian Cancer Panel
Cancer antigen 125	Ovarian Cancer Panel
Human ep growth fact Rec. 2-neu	Ovarian Cancer Panel
Prostate-specific antigen	Prostate Cancer Panel
Free prostate-specific antigen	Prostate Cancer Panel
Complexed prostate-specific antigen	Prostate Cancer Panel
Cocaine	Drug Testing
Diazepam	Drug Testing
Tetrahydrocannabinol	Drug Testing
D-Amphetamine	Drug Testing
Methamphetamine	Drug Testing
Oxazepam	Drug Testing
Nordiazepam	Drug Testing
Temazepam	Drug Testing
Morphine	Drug Testing
Methadone	Drug Testing
MDA	Drug Testing
MDMA	Drug Testing

**Table 2 micromachines-10-00251-t002:** Areas of application, biomarkers and validation studies of bead-based sensors. Work completed here was conducted at various stages of development of the core technology.

Study	Biomarkers	Area	Subjects	Clinical Site	Sponsor
Development of a Lab-on-a-Chip System for Saliva-Based Diagnostics	15 proteins	Cardiac Disease	1000	Baylor College of Medicine-Houston	National Institute of Dental and Craniofacial Research (NIDCR)
Advanced Bio-Nano-Chips for Saliva Based Drug Tests at the Point of Arrest	12 drugs	Drugs of Abuse	240		Home Office-Center of Applied Science and Technology
Next Generation Tools for Onsite Monitoring and Treatment of Drug of Abuse-Dependent Persons	3 drugs	Drugs of Abuse	10 (multiple time points)		National Institute on Drug Abuse (NIDA)
Texas Cancer Diagnostics Pipeline Consortium	4 proteins	Ovarian Cancer	1250	MD Anderson Cancer Clinic-Houston	Cancer Prevention Research Institute of Texas (CPRIT)
	3 proteins	Prostate Cancer	400	UT Health Science Center-San Antonio	
Pilot and Prospective Studies for the Development of the Trauma Chip	5 proteins	Acute Kidney Failure	120	UT Health Science Center-Houston	Texas Emerging Technology Fund
Development of p-BNCs for the Monitoring of Anti-Epilepsy Drugs Levels in Saliva	3 proteins	Epilepsy	100 patients		John S. Dunn Foundation

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
