# Peer review of "Sensors that Learn: The Evolution from Taste Fingerprints to Patterns of Early Disease Detection"

_micromachines, 2019, doi:10.3390/mi10040251_

Round 1

Reviewer 1 Report

In my opinion the idea of a review about sensors paired with machine learning methods nicknamed by the authors as sensors that learn is very interesting. Although I find the manuscript written as a list of accomplishments of the authors rather than a comprehensive review of the field. There are only few systems described but in huge detail. The review lacks conclusions and remarks and guidelines on which readers could build upon and seems to be only praising the developed systems. I do find the systems remarkable, and it is amazing that they were tested with so many real samples I also really liked the initial comparison of Lab-on-a-chip systems with computers but I think the manuscript should be completely rewritten. First of all authors claim that their system from 1998 was the first electronic tongue, how can it be if sensor matrixes coupled with machine learning (more or less IUPAC definition of e-tongues) were proposed a decade earlier! As the topic itself is very interesting especially now in the era of „big data” I recommend for a major revision. Please include work of other groups that work on similar topic, and instead of long descriptions sum up the drawback, indicate the trends and possibilities of growth of the field.

Specific comments:

1. Abstract –what is meant by fluid motivation?

2.  Page 2, sense of taste –the description is very interesting but for comparison with e-tongue also information regarding the mechanics of the system –number of receptors, how the information is coded in the brain etc. would be interesting.

3. Page 3, line 98 –what is meant by molecular reagents?

4. Page 3, line 99 –is the reference 24 correct?

5. Page 3, line 109 and latter –please include references to relevant e-tongue systems from pioneers of the field

6. Although the review is very long the idea behind e-tongues is not really describes please correct

7. Lack of information regarding data analysis methods used in the system

8. Extremely long figure descriptions, and many unnecessary figures (e.g. Fig.4, Fig.5, Fig.6 etc.)

9. Page 6, line 174 –umami is stated as savory shouldn’t it be glutamate?

10. Page 15, Authors include information regarding classification of the grants they received, what is the relevance to the readers?

11. Page 15, 27, Authors describe that their system could help to add new markers to the FDA approved list, it would be interesting to describe what are the problems and how the scientific community can help to add new biomarkers to the list

12. Page 27. Fig 18, Do authors have the rights to use the pictures of equipment used in the drawing?

13. keywords such as Lab-on-chip and point-of-care systems are used throughout the text but there is no place that it is described what is the difference, or if all POC systems have to be LOC (e.g. pregnancy test) it would be interesting to add a table or a decryption characterizing both and underlining differences

Author Response

Rebuttal to REVIEWER #1: "Sensors That Learn: The Evolution from Taste Fingerprints to Patterns of Early Disease Detection” that is targeted for the Special Issue of "Electronic Tongues” for the journal Micromachines.

We would like to thank you for the useful comments to our original submission and recommendations on how to improve it.

Below please find our rebuttal along with a description of how each these suggestions/concerns are addressed.

REVIEWER #1:

Comments and Suggestions for Authors

In my opinion the idea of a review about sensors      paired with machine learning methods nicknamed by the authors as sensors      that learn is very interesting.

We thank Reviewer #1 for expressing this positive remark about the terminology used to describe our sensors. We agree that the combination of topics represents a key development for the area of in vitro diagnostics. We are working hard now to impact patient care through the scaling and distribution of these sensors that learn.

Although I find the manuscript written as a list of      accomplishments of the authors rather than a comprehensive review of the      field. There are only few systems described but in huge detail.

We agree that the original version of the manuscript was too focused on the translational efforts of our laboratory. We accepted the invitation to contribute a review article that summarized the body of work we have completed over that past two decades on this topic. We do understand that there is a huge body of vary deserving work by related groups in this space, but the coverage of a general review was not the focus of our article.

In far too few cases for the lab on a chip community have these medical microdevices matured to the point where clinical translation is possible. Sculley and co-workers have in fact analyzed the NCI grant portfolio from the NIH and shown than less than 2% of their efforts mature to the point that clinical devices are created and about 0.1% evolve to the point where clinical outcomes and impact are documented. The lack of translation is further exacerbated by the recently documented failure of the Silicon Valley start-up, Theranos. The unfortunate hype afforded this effort and the ultimate disappointment has cast a dark shadow on these efforts.

The work described in this effort shows our platform is now moving on a pathway to broad-scale clinical translation. Based on this situation, we feel that it is essential to provide the backdrop of how this technology was developed and how it has evolved from an initial theme of an electronic taste chip to the more recent embodiment of a sensor that learns.

This perspective was not well articulated in the original submitted version of the manuscript. We appreciate the suggestion and have added into the introduction some of these sentiments. The revised draft now includes this message in the introductory section. More importantly, the number of citations has been increased alongside adding in significant new content related to the perspective of research contributions from others in the field.

The review lacks conclusions and remarks and guidelines on which readers could build upon and seems to be only praising the developed systems. I do find the systems remarkable, and it is amazing that they were tested with so many real samples I also really liked the initial comparison of Lab-on-a-chip systems with computers but I think the manuscript should be completely rewritten. First of all authors claim that their system from 1998 was the first electronic tongue, how can it be if sensor matrixes coupled with machine learning (more or less IUPAC definition of e-tongues) were proposed a decade earlier!

We agree fully with these comments also. We have revised this section of the description of the history of the electronic tongues and the electronic taste chip systems to place this work into better historical context. The first electronic tongue is now summarized in page 4 with the new content there located. Also, a significant number of citations (references #9-63) are now included to provide this coverage. The work of David Walt and Dr. Toko, among others, are also included as is more work on the electronic nose activities. While it is not the purpose of the article to summarize the whole field, more broad discussion of this background work is now included to put the work into context.

This discussion of the literature now puts into context the work of the “electronic taste chip” that was the first microbead array work where integrated fluidic elements allow for fluid passage through isolated bead elements that are supported in a MEMS structure. A new conclusion has been crafted to provide more perspective for the field.

As the topic itself is very interesting especially      now in the era of “big data” I recommend for a major revision. Please      include work of other groups that work on similar topic, and instead of      long descriptions sum up the drawback, indicate the trends and      possibilities of growth of the field.

As per this suggestion provided by Reviewer #1, we have implemented the following:

a)    A new section has been written in the discussion that focuses on a perspective of challenges and opportunities that are presented in the path of investigators working in this field.

b)    A new section that includes more detailed mention of other technologies contributing to this field have been added in the introduction section of the manuscript.

c)    A more detailed summary of the contributions of other groups in the big data area as related to chemical sensing is now included on the bottom of page 4.

d)    A new table has been added that summarizes the clinical studies. See page 20

Specific comments:

1.    Abstract –what is meant by fluid motivation?

 The word “motivation” has now been replaced this word with “movement” as it may be more direct and less confusing to the readership.

2. Page 2, sense of taste –the description is very interesting but for comparison with e-tongue also information regarding the mechanics of the system –number of receptors, how the information is coded in the brain etc. would be interesting.

Excellent suggestion! Therefore, a short paragraph was added to include such information.

3. Page 3, line 98 –what is meant by molecular reagents?

We agree with Reviewer 1 this was a rather too general of a descriptor and, thus, was removed from the sentence.

4. Page 3, line 99 –is the reference 24 correct?

Reference 24 was indeed incorrect. The correct reference has now been added. It now reads: “Rakow, N. A.; Suslick, K. S. A colorimetric sensor array for odour visualization. Nature, 2000, 406, 710-713. https://doi.org/10.1038/35021028.”

5. Page 3, line 109 and latter –please include references to relevant e-tongue systems from pioneers of the field.

Another important suggestion by the editor. The revised manuscript now includes a significantly expanded discussion of the relevant background literature. The number of citations has been increased dramatically to 117 references with almost 2/3 of these NOW being directed to the efforts of other researchers. Significant new text has been included to provide perspective comments for the efforts described in our manuscript. On page 4 of the revised document, in addition to Dr. Toko’s work, the extensive list of accomplishments on electrochemical sensors by Dr. Walts’ group is cited and briefly described. Furthermore, in the following page (i.e. page 5) a brief comparison between the two systems (ours vs; Dr. Walt’s) is now included.

6. Although the review is very long the idea behind e-tongues is not really describes please correct.

A new section is now included on pages 3 and 4 of the revised manuscript. Please also see response to #5 as per above.

7. Lack of information regarding data analysis methods used in the system.

A new paragraph and new figure describing data analysis methods in p-BNC bead system are provided now as per reviewer#1 suggestion on pages 15-16, lines 480-499 of the revised manuscript.

8. Extremely long figure descriptions, and many unnecessary figures (e.g. Fig.4, Fig.5, Fig.6 etc.)

As per reviewer’s #1 recommendation, we have trimmed down the descriptors for some of the captions as well as made them more concise. Some figures have indeed been removed but new ones have been added.

9. Page 6, line 174 –umami is stated as savory shouldn’t it be glutamate?

We agree that glutamate is the better descriptor and have changed the text to reflect this perspective.

10. Page 15, Authors include information regarding classification of the grants they received, what is the relevance to the readers?

Based on the reviewer’s #1 recommendation, we have excluded the information reg. classification of the grants.

11. Page 15, 27, Authors describe that their system could help to add new markers to the FDA approved list, it would be interesting to describe what are the problems and how the scientific community can help to add new biomarkers to the list

We now present the challenges in the introduction (page 1, lines 29-41) and initial sections of the discussion (pages 30-32, lines 899-927), as well as suggest a few solutions in the discussion section (page 32, lines 928-940).  

12. Page 27. Fig 18, Do authors have the rights to use the pictures of equipment used in the drawing?

To avoid any confusion, this figure has been removed from the deck.

13. keywords such as Lab-on-chip and point-of-care systems are used throughout the text but there is no place that it is described what is the difference, or if all POC systems have to be LOC (e.g. pregnancy test) it would be interesting to add a table or a decryption characterizing both and underlining differences.

Both terms are decrypted in the introduction section of the revised manuscript.

Further, the following paragraph, in the section entitled “Evolution of the Smart Sensor Platform” seen in page 16, the difference between chip-in-the-lab vs. labs-on-a-chip systems is presented.

Once again, we thank you for your excellent review of the original submission and comments. We hope you find the revised manuscript significantly improved, thanks to your suggestions and comments.

Sincerely,

John T. McDevitt, Ph.D.

Chair, Department Biomaterials

New York University

Reviewer 2 Report

This manuscript has a number of shortcomings that preclude the effective communication and review of literature on the topic of electronic tongue technologies. The following list of weakness in the manuscript are points of serious concern that provide indications of substantial required improvements to bring this paper into an acceptable state of effective communication of recent concepts and principles relating to this area of electronic instruments used in applications for noninvasive early disease detection.

Most of the references are quite dated with relatively few on recent and current concepts; there are only 4 articles cited for 2016 and not any after that year. Thus, very many recent articles and concepts on digital electronic tongue-type devices have been ignored and omitted.

Examples of electronic tongue applications for disease detection (implied by the title) are covered and treated only very generally with no specific medical applications cited.

There is very excessive over use of reprinted materials which demonstrates lack of originality in presenting novel concepts and information from the authors' perspective. This also points to the redundancy of the material presented which are covered by previous reviews. The majority of the 18 figures were from reprinted materials, and figures 8 and 18  presented too general information to be useful.

The nature and style of writing, lacking demonstrated significant mastery of the technologies being reviewed, suggest that the authors have very little experience and personal research to support their knowledge base for writing this review as is evident by the lack of any cited research articles, based on personal research with diagnostic disease-detection instruments.

The authors should strongly consider the writing of reviews based on experience gained from substantial personal research. The reiteration of concepts presented in outdated older papers by other authors does not provide the basis for a contribution that is either current or novel in presenting cutting-edge concepts in the area of disease diagnostic technologies.

Author Response

Rebuttal to REVIEWER #2: "Sensors That Learn: The Evolution from Taste Fingerprints to Patterns of Early Disease Detection” that is targeted for the Special Issue of "Electronic Tongues” for the journal Micromachines.

Comments and Suggestions for Authors

This manuscript has a number of shortcomings that preclude the effective communication and review of literature on the topic of electronic tongue technologies. The following list of weakness in the manuscript are points of serious concern that provide indications of substantial required improvements to bring this paper into an acceptable state of effective communication of recent concepts and principles relating to this area of electronic instruments used in applications for noninvasive early disease detection.

Most of      the references are quite dated with relatively few on recent and current      concepts; there are only 4 articles cited for 2016 and not any after that      year. Thus, very many recent articles and concepts on digital electronic      tongue-type devices have been ignored and omitted.

Examples      of electronic tongue applications for disease detection (implied by the      title) are covered and treated only very generally with no specific      medical applications cited.

There is very excessive over use of      reprinted materials which demonstrates lack of originality in presenting      novel concepts and information from the authors' perspective. This also      points to the redundancy of the material presented which are covered by      previous reviews. The majority of the 18 figures were from reprinted      materials, and figures 8 and 18  presented too general information to      be useful.

The nature and style of writing,      lacking demonstrated significant mastery of the technologies being      reviewed, suggest that the authors have very little experience and      personal research to support their knowledge base for writing this review      as is evident by the lack of any cited research articles, based on      personal research with diagnostic disease-detection instruments.

5.    The authors should strongly consider the writing of reviews based on experience gained from substantial personal research. The reiteration of concepts presented in outdated older papers by other authors does not provide the basis for a contribution that is either current or novel in presenting cutting-edge concepts in the area of disease diagnostic technologies.

REBUTTAL

We appreciate the constructive criticism from Reviewer 2 and have made extensive revisions along the lines of these comments.

A list of actions taken in response to each of the specific comments follows.

1.    Most of the references are quite dated with relatively few on recent and current concepts; there are only 4 articles cited for 2016 and not any after that year. Thus, very many recent articles and concepts on digital electronic tongue-type devices have been ignored and omitted.

We have revised the section of the description of the history of the electronic tongues and the electronic taste chip systems to place this work into better historical context. The first electronic tongue are now summarized in page 4 with the new content there located. Also, citations 9-63 are included to provide this coverage. The work of David Walt and Dr. Toko, among others, are also included as is more work on the electronic nose activities. While it is not the purpose of the article to summarize the whole field, more broad discussion of this background work is now included to put the work into context. This discussion of the literature now puts into context the work of the “electronic taste chip” that was the first microbead array work where integrated fluidic elements allow for fluid passage through isolated bead elements that are supported in a MEMS structure.

Examples of electronic tongue      applications for disease detection (implied by the title) are covered and      treated only very generally with no specific medical applications cited.

This is a good suggestion. A new table has been added to the revised manuscript to better summarize the clinical studies (page 20, Table 2). The introduction (pages 1 and 2) and the section dedicated to electronic nose and electronic tongue (pages 3-5) have been rewritten to better emphasize the importance of the clinical translation. Furthermore, a new manuscript summary section (Page 1, abstract) has been drafted, as well as a new discussion that now also includes text and Figure 25 (pages 30-31), both related to the translation theme.

There is very excessive over use of      reprinted materials which demonstrates lack of originality in presenting      novel concepts and information from the authors' perspective. This also      points to the redundancy of the material presented which are covered by      previous reviews. The majority of the 18 figures were from reprinted      materials, and figures 8 and 18  presented too general information to      be useful.

To address this concern, we have removed a number of the original figures. We have completely rewritten the paper to add in additional perspective. We have added in new diagnostic information not published previously (i.e. for allergy testing). We have added in a number of new figures not published previously.

These include:

Figure 4. A broad range of applications have been demonstrated for the p-BNC sensor approach. The various test sectors are summarized in this illustration/page 9

Figure 5. This illustration shows dose response curves for cardiac assays as developed on a partially integrated p-BNC sensing platform/page10

Figure 6: Precision data that was acquired for the cardiac biomarker c-Tn-I on a partially integrated p-BNC platform/page 11.

Figure 7: Calibration curves are shown for the following cases: (A) CA125, (B) HE4, (C) MMP-7, and (D) CA72-4./page12.

Figure 12. Summary of the bead analysis algorithm shown for one bead in the array/page 16

            Figure 13. The evolution of the instrumentation/page 17.

Table 2 Areas of application, biomarkers and validation studies of bead-based p-BNC/page 20.

Figure 18. (A) Allergy testing/page 24

Figure 24: A) The two valleys of “death” that medical device manufactures face and B) the E5D bridge that connects academic findings with their widespread usage at the clinical setting/page 31

Figure 25. The current SensoDx instrumentation/page 32

4.    The nature and style of writing, lacking demonstrated significant mastery of the technologies being reviewed, suggest that the authors have very little experience and personal research to support their knowledge base for writing this review as is evident by the lack of any cited research articles, based on personal research with diagnostic disease-detection instruments.

This perspective was not well articulated in the original submitted version of the manuscript which was submitted under deadlines. We do appreciate the comments from the reviewer and have added into the introduction new content that provides more perspective of the field and coverage of the unique translational aspects of this approach as it relates to complex fluid analysis. The revised manuscript now includes a significantly expanded discussion of the relevant background literature. A large number of new citations have been included alongside text describing these efforts and how they relate back to the topic of this paper. The number of citations has been increased dramatically with about 2/3 of these NOW being directed to the efforts of other researchers. Significant new text has been included to provide perspective comments for the efforts described in our manuscript.

The revised draft now includes this perspective in multiple locations as collectively described in the answers to concerns 1-3 as per above.

We regret that we did not make these points more clearly in the original submission.

It should be noted that we accepted the invitation to contribute a review article that summarized the body of work we have completed over that past two decades on this topic. We do understand that there is a huge body of vary deserving work by related groups in this space, but the coverage of a general review was not the focus of our article.

In far too few cases for the lab on a chip community have these medical microdevices matured to the point where clinical translation is possible. Sculley and co-workers have in fact analyzed the NCI grant portfolio from the NIH and shown than less than 2% of their efforts mature to the point that clinical devices are created and about 0.1% evolve to the point where clinical outcomes and impact are documented. The lack of translation is further exacerbated by the recently documented failure of the Silicon Valley start-up, Theranos. The unfortunate hype afforded this effort and the ultimate disappointment has cast a dark shadow on these efforts.

The work described in this effort shows our platform is now moving on a pathway to broad-scale clinical translation. Based on this situation, we feel that it is essential to provide the backdrop of how this technology was developed and how it has evolved from an initial theme of an electronic taste chip to the more recent embodiment of a sensor that learns.

5.    The authors should strongly consider the writing of reviews based on experience gained from substantial personal research. The reiteration of concepts presented in outdated older papers by other authors does not provide the basis for a contribution that is either current or novel in presenting cutting-edge concepts in the area of disease diagnostic technologies.

We fully agree with this sentiment and have made significant changes in a revised submission. The changes are covered in the 4 sections listed above.

One again, we would like to thank Reviewer #2 for the useful comments to our original submission and recommendations on how to improve it.

Sincerely,

John T. McDevitt, Ph.D.

Chair, Department Biomaterials

New York University

Round 2

Reviewer 1 Report

I appreciate the corrections made by the authors, but one of my suggestions was that the manusript is too long, and instead of shortening it grew even bigger (36 to 42 pages). I fully understand that the authors did a huge number of devices and I do find them interesting, but a review like that should not describe each device in detail. I still think that general trends and guidelines for readers should be provided and interested readers can find the details of construction in the cited literature. Shortening the work at least by half and providing some more conclussions and disscussions  regarding the described topics would greatly improved this work and make it a guidelike for other reserachers to follow.